# Training Energy-Based Normalizing Flow with Score-Matching Objectives

**Chen-Hao Chao[1], Wei-Fang Sun[1,2], Yen-Chang Hsu[3], Zsolt Kira[4], and Chun-Yi Lee[1]***

[1] Elsa Lab, National Tsing Hua University, Hsinchu City, Taiwan
[2] NVIDIA AI Technology Center, NVIDIA Corporation, Santa Clara, CA, USA
[3] Samsung Research America, Mountain View, CA, USA
[4] Georgia Institute of Technology, Atlanta, GA, USA

## Abstract

In this paper, we establish a connection between the parameterization of flow-based and energy-based generative models, and present a new flow-based modeling approach called energy-based normalizing flow (EBFlow). We demonstrate that by optimizing EBFlow with score-matching objectives, the computation of Jacobian determinants for linear transformations can be entirely bypassed. This feature enables the use of arbitrary linear layers in the construction of flow-based models without increasing the computational time complexity of each training iteration from $\mathcal{O}(D^2L)$ to $\mathcal{O}(D^3L)$ for an $L$-layered model that accepts $D$-dimensional inputs. This makes the training of EBFlow more efficient than the commonly-adopted maximum likelihood training method. In addition to the reduction in runtime, we enhance the training stability and empirical performance of EBFlow through a number of techniques developed based on our analysis of the score-matching methods. The experimental results demonstrate that our approach achieves a significant speedup compared to maximum likelihood estimation while outperforming prior methods with a noticeable margin in terms of negative log-likelihood (NLL).

## 1 Introduction

Parameter estimation for probability density functions (pdf) has been a major interest in the research fields of machine learning and statistics. Given a $D$-dimensional random data vector $\mathbf{x} \in \mathbb{R}^D$, the goal of such a task is to estimate the true pdf $p_{\mathbf{x}}(\cdot)$ of $\mathbf{x}$ with a function $p(\cdot\,;\theta)$ parameterized by $\theta$. In the studies of unsupervised learning, flow-based modeling methods (e.g., [1–4]) are commonly-adopted for estimating $p_{\mathbf{x}}$ due to their expressiveness and broad applicability in generative tasks.

Flow-based models represent $p(\cdot\,;\theta)$ using a sequence of invertible transformations based on the change of variable theorem, through which the intermediate unnormalized densities are re-normalized by multiplying the Jacobian determinant associated with each transformation. In maximum likelihood estimation, however, the explicit computation of the normalizing term may pose computational challenges for model architectures that use linear transformations, such as convolutions [4, 5] and fully-connected layers [6, 7]. To address this issue, several methods have been proposed in the recent literature, which includes constructing linear transformations with special structures [8–12] and exploiting special optimization processes [7]. Despite their success in reducing the training complexity, these methods either require additional constraints on the linear transformations or biased estimation on the gradients of the objective.

Motivated by the limitations of the previous studies, this paper introduces an approach that reinterprets flow-based models as energy-based models [13], and leverages score-matching methods [14–17] to

---

*Corresponding author. Email: `cylee@cs.nthu.edu.tw`

37th Conference on Neural Information Processing Systems (NeurIPS 2023).

optimize $p(\cdot\,;\theta)$ according to the Fisher divergence [14, 18] between $p_{\mathbf{x}}(\cdot)$ and $p(\cdot\,;\theta)$. The proposed method avoids the computation of the Jacobian determinants of linear layers during training, and reduces the asymptotic computational complexity of each training iteration from $\mathcal{O}(D^3 L)$ to $\mathcal{O}(D^2 L)$ for an $L$-layered model. Our experimental results demonstrate that this approach significantly improves the training efficiency as compared to maximum likelihood training. In addition, we investigate a theoretical property of Fisher divergence with respect to latent variables, and propose a Match-after-Preprocessing (MaP) technique to enhance the training stability of score-matching methods. Finally, our comparison on the MNIST dataset [19] reveals that the proposed method exhibit significant improvements in comparison to our baseline methods presented in [17] and [7] in terms of negative log likelihood (NLL).

## 2   Background

In this section, we discuss the parameterization of probability density functions in flow-based and energy-based modeling methods, and offer a number of commonly-used training methods for them.

### 2.1   Flow-based Models

Flow-based models describe $p_{\mathbf{x}}(\cdot)$ using a prior distribution $p_{\mathbf{u}}(\cdot)$ of a latent variable $\mathbf{u} \in \mathbb{R}^D$ and an invertible function $g = g_L \circ \cdots \circ g_1$, where $g_i(\cdot\,;\theta) : \mathbb{R}^D \to \mathbb{R}^D$, $\forall i \in \{1, \cdots, L\}$ and is usually modeled as a neural network with $L$ layers. Based on the change of variable theorem and the distributive property of the determinant operation $\det(\cdot)$, $p(\cdot\,;\theta)$ can be described as follows:

$$p(\boldsymbol{x};\theta) = p_{\mathbf{u}}\left(g(\boldsymbol{x};\theta)\right) |\det(\mathbf{J}_g(\boldsymbol{x};\theta))| = p_{\mathbf{u}}\left(g(\boldsymbol{x};\theta)\right) \prod_{i=1}^{L} |\det(\mathbf{J}_{g_i}(\boldsymbol{x}_{i-1};\theta))|, \qquad (1)$$

where $\boldsymbol{x}_i = g_i \circ \cdots \circ g_1(\boldsymbol{x};\theta)$, $\boldsymbol{x}_0 = \boldsymbol{x}$, $\mathbf{J}_g(\boldsymbol{x};\theta) = \frac{\partial}{\partial \boldsymbol{x}} g(\boldsymbol{x};\theta)$ represents the Jacobian of $g$ with respect to $\boldsymbol{x}$, and $\mathbf{J}_{g_i}(\boldsymbol{x}_{i-1};\theta) = \frac{\partial}{\partial \boldsymbol{x}_{i-1}} g_i(\boldsymbol{x}_{i-1};\theta)$ represents the Jacobian of the $i$-th layer of $g$ with respect to $\boldsymbol{x}_{i-1}$. This work concentrates on model architectures employing *linear flows* [20] to design the function $g$. These model architectures primarily utilize linear transformations to extract crucial feature representations, while also accommodating non-linear transformations that enable efficient Jacobian determinant computation. Specifically, let $\mathcal{S}_l$ be the set of linear transformations in $g$, and $\mathcal{S}_n = \{g_i \,|\, i \in \{1, \cdots, L\}\} \setminus \mathcal{S}_l$ be the set of non-linear transformations. The general assumption of these model architectures is that $\prod_{i=1}^{L} |\det(\mathbf{J}_{g_i})|$ in Eq. (1) can be decomposed as $\prod_{g_i \in \mathcal{S}_n} |\det(\mathbf{J}_{g_i})| \prod_{g_i \in \mathcal{S}_l} |\det(\mathbf{J}_{g_i})|$, where $\prod_{g_i \in \mathcal{S}_n} |\det(\mathbf{J}_{g_i})|$ and $\prod_{g_i \in \mathcal{S}_l} |\det(\mathbf{J}_{g_i})|$ can be calculated within the complexity of $\mathcal{O}(D^2 L)$ and $\mathcal{O}(D^3 L)$, respectively. Previous implementations of such model architectures include Generative Flows (Glow) [4], Neural Spline Flows (NSF) [5], and the independent component analysis (ICA) models presented in [6, 7].

Given the parameterization of $p(\cdot\,;\theta)$, a commonly used approach for optimizing $\theta$ is maximum likelihood (ML) estimation, which involves minimizing the Kullback-Leibler (KL) divergence $\mathbb{D}_{\mathrm{KL}}\left[p_{\mathbf{x}}(\boldsymbol{x}) \| p(\boldsymbol{x};\theta)\right] = \mathbb{E}_{p_{\mathbf{x}}(\boldsymbol{x})}\left[\log \frac{p_{\mathbf{x}}(\boldsymbol{x})}{p(\boldsymbol{x};\theta)}\right]$ between the true density $p_{\mathbf{x}}(\boldsymbol{x})$ and the parameterized density $p(\boldsymbol{x};\theta)$. The ML objective $\mathcal{L}_{\mathrm{ML}}(\theta)$ is derived by removing the constant term $\mathbb{E}_{p_{\mathbf{x}}(\boldsymbol{x})}\left[\log p_{\mathbf{x}}(\boldsymbol{x})\right]$ with respect to $\theta$ from $\mathbb{D}_{\mathrm{KL}}\left[p_{\mathbf{x}}(\boldsymbol{x}) \| p(\boldsymbol{x};\theta)\right]$, and can be expressed as follows:

$$\mathcal{L}_{\mathrm{ML}}(\theta) = \mathbb{E}_{p_{\mathbf{x}}(\boldsymbol{x})}\left[-\log p(\boldsymbol{x};\theta)\right]. \qquad (2)$$

The ML objective explicitly evaluates $p(\boldsymbol{x};\theta)$, which involves the calculation of the Jacobian determinant of the layers in $\mathcal{S}_l$. This indicates that certain model architectures containing convolutional [4, 5] or fully-connected layers [6, 7] may encounter training inefficiency due to the $\mathcal{O}(D^3 L)$ cost of evaluating $\prod_{g_i \in \mathcal{S}_l} |\det(\mathbf{J}_{g_i})|$. Although a number of alternative methods discussed in Section 3 can be adopted to reduce their computational cost, they either require additional constraints on the linear transformation or biased estimation on the gradients of the ML objective.

### 2.2   Energy-based Models

Energy-based models are formulated based on a Boltzmann distribution, which is expressed as the ratio of an unnormalized density function to an input-independent normalizing constant. Specif-

ically, given a scalar-valued energy function $E(\cdot\,;\theta) : \mathbb{R}^D \to \mathbb{R}$, the unnormalized density function is defined as $\exp\left(-E(\boldsymbol{x};\theta)\right)$, and the normalizing constant $Z(\theta)$ is defined as the integration $\int_{\boldsymbol{x}\in\mathbb{R}^D} \exp\left(-E(\boldsymbol{x};\theta)\right) d\boldsymbol{x}$. The parameterization of $p(\cdot\,;\theta)$ is presented in the following equation:

$$p(\boldsymbol{x};\theta) = \exp\left(-E(\boldsymbol{x};\theta)\right) Z^{-1}(\theta). \tag{3}$$

Optimizing $p(\cdot\,;\theta)$ in Eq. (3) through directly evaluating $\mathcal{L}_{\mathrm{ML}}$ in Eq. (2) is computationally infeasible, since the computation requires explicitly calculating the intractable normalizing constant $Z(\theta)$. To address this issue, a widely-used technique [13] is to reformulate $\frac{\partial}{\partial\theta}\mathcal{L}_{\mathrm{ML}}(\theta)$ as its sampling-based variant $\frac{\partial}{\partial\theta}\mathcal{L}_{\mathrm{SML}}(\theta)$, which is expressed as follows:

$$\mathcal{L}_{\mathrm{SML}}(\theta) = \mathbb{E}_{p_{\mathbf{x}}(\boldsymbol{x})}\left[E(\boldsymbol{x};\theta)\right] - \mathbb{E}_{\mathrm{sg}(p(\boldsymbol{x};\theta))}\left[E(\boldsymbol{x};\theta)\right], \tag{4}$$

where $\mathrm{sg}(\cdot)$ indicates the stop-gradient operator. Despite the fact that Eq. (4) prevents the calculation of $Z(\theta)$, sampling from $p(\cdot\,;\theta)$ typically requires running a Markov Chain Monte Carlo (MCMC) process (e.g., [21, 22]) until convergence, which can still be computationally expensive as it involves evaluating the gradients of the energy function numerous times. Although several approaches [23, 24] were proposed to mitigate the high computational costs involved in performing an MCMC process, these approaches make use of approximations, which often cause training instabilities in high-dimensional contexts [25].

Another line of researches proposed to optimize $p(\cdot\,;\theta)$ through minimizing the Fisher divergence $\mathbb{D}_{\mathrm{F}}\left[p_{\mathbf{x}}(\boldsymbol{x})\|p(\boldsymbol{x};\theta)\right] = \mathbb{E}_{p_{\mathbf{x}}(\boldsymbol{x})}\left[\frac{1}{2}\|\frac{\partial}{\partial\boldsymbol{x}}\log\left(\frac{p_{\mathbf{x}}(\boldsymbol{x})}{p(\boldsymbol{x};\theta)}\right)\|^2\right]$ between $p_{\mathbf{x}}(\boldsymbol{x})$ and $p(\boldsymbol{x};\theta)$ using the score-matching (SM) objective $\mathcal{L}_{\mathrm{SM}}(\theta) = \mathbb{E}_{p_{\mathbf{x}}(\boldsymbol{x})}\left[\frac{1}{2}\|\frac{\partial}{\partial\boldsymbol{x}}E(\boldsymbol{x};\theta)\|^2 - \mathrm{Tr}(\frac{\partial^2}{\partial\boldsymbol{x}^2}E(\boldsymbol{x};\theta))\right]$ [14] to avoid the explicit calculation of $Z(\theta)$ as well as the sampling process required in Eq. (4). Several computationally efficient variants of $\mathcal{L}_{\mathrm{SM}}$, including sliced score matching (SSM) [16], finite difference sliced score matching (FDSSM) [17], and denoising score matching (DSM) [15], have been proposed.

SSM is derived directly based on $\mathcal{L}_{\mathrm{SM}}$ with an unbiased Hutchinson's trace estimator [26]. Given a random projection vector $\mathbf{v} \in \mathbb{R}^D$ drawn from $p_{\mathbf{v}}$ and satisfying $\mathbb{E}_{p_{\mathbf{v}}(\boldsymbol{v})}[\boldsymbol{v}^T\boldsymbol{v}] = \boldsymbol{I}$, the objective function denoted as $\mathcal{L}_{\mathrm{SSM}}$, is defined as follows:

$$\mathcal{L}_{\mathrm{SSM}}(\theta) = \frac{1}{2}\mathbb{E}_{p_{\mathbf{x}}(\boldsymbol{x})}\left[\left\|\frac{\partial E(\boldsymbol{x};\theta)}{\partial\boldsymbol{x}}\right\|^2\right] - \mathbb{E}_{p_{\mathbf{x}}(\boldsymbol{x})p_{\mathbf{v}}(\boldsymbol{v})}\left[\boldsymbol{v}^T\frac{\partial^2 E(\boldsymbol{x};\theta)}{\partial\boldsymbol{x}^2}\boldsymbol{v}\right]. \tag{5}$$

FDSSM is a parallelizable variant of $\mathcal{L}_{\mathrm{SSM}}$ that adopts the finite difference method [27] to approximate the gradient operations in the objective. Given a uniformly distributed random vector $\varepsilon$, it accelerates the calculation by simultaneously forward passing $E(\boldsymbol{x};\theta)$, $E(\boldsymbol{x}+\varepsilon;\theta)$, and $E(\boldsymbol{x}-\varepsilon;\theta)$ as follows:

$$\begin{aligned}\mathcal{L}_{\mathrm{FDSSM}}(\theta) = {} & 2\mathbb{E}_{p_{\mathbf{x}}(\boldsymbol{x})}\left[E(\boldsymbol{x};\theta)\right] - \mathbb{E}_{p_{\mathbf{x}}(\boldsymbol{x})p_{\xi}(\varepsilon)}\left[E(\boldsymbol{x}+\varepsilon;\theta) + E(\boldsymbol{x}-\varepsilon;\theta)\right] \\ & + \frac{1}{8}\mathbb{E}_{p_{\mathbf{x}}(\boldsymbol{x})p_{\xi}(\varepsilon)}\left[\left(E(\boldsymbol{x}+\varepsilon;\theta) - E(\boldsymbol{x}-\varepsilon;\theta)\right)^2\right],\end{aligned} \tag{6}$$

where $p_{\xi}(\varepsilon) = \mathcal{U}(\varepsilon \in \mathbb{R}^D|\,\|\varepsilon\| = \xi)$, and $\xi$ is a hyper-parameter that usually assumes a small value.

DSM approximates the true pdf through a surrogate that is constructed using the Parzen density estimator $p_{\sigma}(\tilde{\boldsymbol{x}})$ [28]. The approximated target $p_{\sigma}(\tilde{\boldsymbol{x}}) = \int_{\boldsymbol{x}\in\mathbb{R}^D} p_{\sigma}(\tilde{\boldsymbol{x}}|\boldsymbol{x})p_{\mathbf{x}}(\boldsymbol{x})d\boldsymbol{x}$ is defined based on an isotropic Gaussian kernel $p_{\sigma}(\tilde{\boldsymbol{x}}|\boldsymbol{x}) = \mathcal{N}(\tilde{\boldsymbol{x}}|\boldsymbol{x},\sigma^2\boldsymbol{I})$ with a variance $\sigma^2$. The objective $\mathcal{L}_{\mathrm{DSM}}$, which excludes the Hessian term in $\mathcal{L}_{\mathrm{SSM}}$, is written as follows:

$$\mathcal{L}_{\mathrm{DSM}}(\theta) = \mathbb{E}_{p_{\mathbf{x}}(\boldsymbol{x})p_{\sigma}(\tilde{\boldsymbol{x}}|\boldsymbol{x})}\left[\frac{1}{2}\left\|\frac{\partial E(\tilde{\boldsymbol{x}};\theta)}{\partial\tilde{\boldsymbol{x}}} + \frac{\boldsymbol{x}-\tilde{\boldsymbol{x}}}{\sigma^2}\right\|^2\right]. \tag{7}$$

To conclude, $\mathcal{L}_{\mathrm{SSM}}$ is an unbiased objective that satisfies $\frac{\partial}{\partial\theta}\mathcal{L}_{\mathrm{SSM}}(\theta) = \frac{\partial}{\partial\theta}\mathbb{D}_{\mathrm{F}}\left[p_{\mathbf{x}}(\boldsymbol{x})\|p(\boldsymbol{x};\theta)\right]$ [16], while $\mathcal{L}_{\mathrm{FDSSM}}$ and $\mathcal{L}_{\mathrm{DSM}}$ require careful selection of hyper-parameters $\xi$ and $\sigma$, since $\frac{\partial}{\partial\theta}\mathcal{L}_{\mathrm{FDSSM}}(\theta) = \frac{\partial}{\partial\theta}(\mathbb{D}_{\mathrm{F}}\left[p_{\mathbf{x}}(\boldsymbol{x})\|p(\boldsymbol{x};\theta)\right] + o\left(\xi\right))$ [17] contains an approximation error $o\left(\xi\right)$, and $p_{\sigma}$ in $\frac{\partial}{\partial\theta}\mathcal{L}_{\mathrm{DSM}}(\theta) = \frac{\partial}{\partial\theta}\mathbb{D}_{\mathrm{F}}\left[p_{\sigma}(\tilde{\boldsymbol{x}})\|p(\tilde{\boldsymbol{x}};\theta)\right]$ may bear resemblance to $p_{\mathbf{x}}$ only for small $\sigma$ [15].

# 3 Related Works

## 3.1 Accelerating Maximum Likelihood Training of Flow-based Models

A key focus in the field of flow-based modeling is to reduce the computational expense associated with evaluating the ML objective [7–12, 29]. These acceleration methods can be classified into two categories based on their underlying mechanisms.

**Specially Designed Linear Transformations.** A majority of the existing works [8–12, 29] have attempted to accelerate the computation of Jacobian determinants in the ML objective by exploiting linear transformations with special structures. For example, the authors in [8] proposed to constrain the weights in linear layers as lower triangular matrices to speed up training. The authors in [9, 10] proposed to adopt convolutional layers with masked kernels to accelerate the computation of Jacobian determinants. The authors in [29] leveraged orthogonal transformations to bypass the direct computation of Jacobian determinants. More recently, the authors in [12] proposed to utilize linear operations with special *butterfly* structures [30] to reduce the cost of calculating the determinants. Although these techniques avoid the $\mathcal{O}(D^3L)$ computation, they impose restrictions on the learnable transformations, which potentially limits their capacity to capture complex feature representations, as discussed in [7, 31, 32]. Our experimental findings presented in Appendix A.5 support this concept, demonstrating that flow-based models with unconstrained linear layers outperform those with linear layers restricted by lower / upper triangular weight matrices [8] or those using lower–upper (LU) decomposition [4].

**Specially Designed Optimization Process.** To address the aforementioned restrictions, a recent study [7] proposed the relative gradient method for optimizing flow-based models with arbitrary linear transformations. In this method, the gradients of the ML objective are converted into their relative gradients by multiplying themselves with $\boldsymbol{W}^T\boldsymbol{W}$, where $\boldsymbol{W} \in \mathbb{R}^{D \times D}$ represents the weight matrix in a linear transformation. Since $\frac{\partial}{\partial \boldsymbol{W}} \log |\det(\boldsymbol{W})| \boldsymbol{W}^T\boldsymbol{W} = \boldsymbol{W}$, evaluating relative gradients is more computationally efficient than calculating the standard gradients according to $\frac{\partial}{\partial \boldsymbol{W}} \log |\det(\boldsymbol{W})| = (\boldsymbol{W}^T)^{-1}$. While this method reduces the training time complexity from $\mathcal{O}(D^3L)$ to $\mathcal{O}(D^2L)$, a significant downside to this approach is that it introduces approximation errors with a magnitude of $o(\boldsymbol{W})$, which can escalate relative to the weight matrix values.

## 3.2 Training Flow-based Models with Score-Matching Objectives

The pioneering study [14] is the earliest attempt to train flow-based models by minimizing the SM objective. Their results demonstrate that models trained using the SM loss are able to achieve comparable or even better performance to those trained with the ML objective in a low-dimensional experimental setup. More recently, the authors in [16] and [17] proposed two efficient variants of the SM loss, i.e., the SSM and FDSSM objectives, respectively. They demonstrated that these loss functions can be used to train a non-linear independent component estimation (NICE) [1] model on high-dimensional tasks. While the training approaches of these works bear resemblance to ours, our proposed method places greater emphasis on training efficiency. Specifically, they directly implemented the energy function $E(\boldsymbol{x}; \theta)$ in the score-matching objectives as $-\log p(\boldsymbol{x}; \theta)$, resulting in a significantly higher computational cost compared to our method introduced in Section 4. In Section 5, we further demonstrate that the models trained with the methods in [16, 17] yield less satisfactory results in comparison to our approach.

# 4 Methodology

In this section, we introduce a new framework for reducing the training cost of flow-based models with linear transformations, and discuss a number of training techniques for enhancing its performance.

## 4.1 Energy-Based Normalizing Flow

Instead of applying architectural constraints to reduce computational time complexity, we achieve the same goal through adopting the training objectives of energy-based models. We name this approach as Energy-Based Normalizing Flow (EBFlow). A key observation is that the parametric density function of a flow-based model can be reinterpreted as that of an energy-based model through identifying

the input-independent multipliers in $p(\,\cdot\,;\theta)$. Specifically, $p(\,\cdot\,;\theta)$ can be explicitly factorized into an unnormalized density and a corresponding normalizing term as follows:

$$p(\boldsymbol{x};\theta) = p_{\mathbf{u}}\left(g(\boldsymbol{x};\theta)\right) \prod_{i=1}^{L} \left|\det\left(\mathbf{J}_{g_i}(\boldsymbol{x}_{i-1};\theta)\right)\right|$$

$$= \underbrace{p_{\mathbf{u}}\left(g(\boldsymbol{x};\theta)\right) \prod_{g_i \in \mathcal{S}_n} \left|\det\left(\mathbf{J}_{g_i}(\boldsymbol{x}_{i-1};\theta)\right)\right|}_{\text{Unnormalized Density}} \underbrace{\prod_{g_i \in \mathcal{S}_l} \left|\det(\mathbf{J}_{g_i}(\theta))\right|}_{\text{Norm. Const.}} \triangleq \underbrace{\exp\left(-E(\boldsymbol{x};\theta)\right)}_{\text{Unnormalized Density}} \underbrace{Z^{-1}(\theta)}_{\text{Norm. Const.}}$$

(8)

where the energy function $E(\,\cdot\,;\theta)$ and the normalizing constant $Z^{-1}(\theta)$ are selected as follows:

$$E(\boldsymbol{x};\theta) \triangleq -\log\left(p_{\mathbf{u}}\left(g(\boldsymbol{x};\theta)\right) \prod_{g_i \in \mathcal{S}_n} \left|\det(\mathbf{J}_{g_i}(\boldsymbol{x}_{i-1};\theta))\right|\right), Z^{-1}(\theta) = \prod_{g_i \in \mathcal{S}_l} \left|\det(\mathbf{J}_{g_i}(\theta))\right|. \quad (9)$$

The detailed derivations of Eqs. (8) and (9) are elaborated in Lemma A.11 of Section A.1.2. By isolating the computationally expensive term in $p(\,\cdot\,;\theta)$ as the normalizing constant $Z(\theta)$, the parametric pdf defined in Eqs. (8) and (9) becomes suitable for the training methods of energy-based models. In the subsequent paragraphs, we discuss the training, inference, and convergence property of EBFlow.

**Training Cost.** Based on the definition in Eqs. (8) and (9), the score-matching objectives specified in Eqs. (5)-(7) can be adopted to prevent the Jacobian determinant calculation for the elements in $\mathcal{S}_l$. As a result, the training complexity can be significantly reduced to $\mathcal{O}(D^2 L)$, as the $\mathcal{O}(D^3 L)$ calculation of $Z(\theta)$ is completely avoided. Such a design allows the use of arbitrary linear transformations in the construction of a flow-based model without posing computational challenge during the training process. This feature is crucial to the architectural flexibility of a flow-based model. For example, fully-connected layers and convolutional layers with arbitrary padding and striding strategies can be employed in EBFlow without increasing the training complexity. EBFlow thus exhibits an enhanced flexibility in comparison to the related works that exploit specially designed linear transformations.

**Inference Cost.** Although the computational cost of evaluating the exact Jacobian determinants of the elements in $\mathcal{S}_l$ still requires $\mathcal{O}(D^3 L)$ time, these operations can be computed only once after training and reused for subsequent inferences, since $Z(\theta)$ is a constant as long as $\theta$ is fixed. In cases where $D$ is extremely large and $Z(\theta)$ cannot be explicitly calculated, stochastic estimators such as the importance sampling techniques (e.g., [33, 34]) can be used as an alternative to approximate $Z(\theta)$. We provide a brief discussion of such a scenario in Appendix A.3.

**Asymptotic Convergence Property.** Similar to maximum likelihood training, score-matching methods that minimize Fisher divergence have theoretical guarantees on their *consistency* [14, 16]. This property is essential in ensuring the convergence accuracy of the parameters. Let $N$ be the number of independent and identically distributed (i.i.d.) samples drawn from $p_{\mathbf{x}}$ to approximate the expectation in the SM objective. In addition, assume that there exists a set of optimal parameters $\theta^*$ such that $p(\boldsymbol{x};\theta^*) = p_{\mathbf{x}}(\boldsymbol{x})$. Under the regularity conditions (i.e., Assumptions A.1-A.7 shown in Appendix A.1.1), *consistency* guarantees that the parameters $\theta_N$ minimizing the SM loss converges (in probability) to its optimal value $\theta^*$ when $N \to \infty$, i.e., $\theta_N \xrightarrow{p} \theta^*$ as $N \to \infty$. In Appendix A.1.1, we provide a formal description of this property based on [16] and derive the sufficient condition for $g$ and $p_{\mathbf{u}}$ to satisfy the regularity conditions (i.e., Proposition A.10).

## 4.2 Techniques for Enhancing the Training of EBFlow

As revealed in the recent studies [16, 17], training flow-based models with score-matching objectives is challenging as the training process is numerically unstable and usually exhibits significant variances. To address these issues, we propose to adopt two techniques: match after preprocessing (MaP) and exponential moving average (EMA), which are particularly effective in dealing with the above issues according to our ablation analysis in Section 5.3.

**MaP.** Score-matching methods rely on the score function $-\frac{\partial}{\partial \boldsymbol{x}} E(\boldsymbol{x};\theta)$ to match $\frac{\partial}{\partial \boldsymbol{x}} \log p_{\mathbf{x}}(\boldsymbol{x})$, which requires backward propagation through each layer in $g$. This indicates that the training process could be numerically sensitive to the derivatives of $g$. For instance, logit pre-processing layers commonly used in flow-based models (e.g., [1, 4, 5, 7, 8, 35]) exhibit extremely large derivatives near 0 and 1, which might exacerbate the above issue. To address this problem, we propose to

exclude the numerically sensitive layer(s) from the model and match the pdf of the pre-processed variable during training. Specifically, let $\mathbf{x}_k \triangleq g_k \circ \cdots \circ g_1(\mathbf{x})$ be the pre-processed variable, where $k$ represents the index of the numerically sensitive layer. This method aims to optimize a parameterized pdf $p_k(\cdot\,;\theta) \triangleq p_{\mathbf{u}}(g_L \circ \cdots \circ g_{k+1}(\cdot\,;\theta)) \prod_{i=k+1}^{L} |\det(\mathbf{J}_{g_i})|$ that excludes $(g_k, \cdots, g_1)$ through minimizing the Fisher divergence between the pdf $p_{\mathbf{x}_k}(\cdot)$ of $\mathbf{x}_k$ and $p_k(\cdot\,;\theta)$ by considering the (local) behavior of $\mathbb{D}_{\mathrm{F}}$, as presented in Proposition 4.1.

**Proposition 4.1.** *Let $p_{\mathbf{x}_j}$ be the pdf of the latent variable of $\mathbf{x}_j \triangleq g_j \circ \cdots \circ g_1(\mathbf{x})$ indexed by $j$. In addition, let $p_j(\cdot)$ be a pdf modeled as $p_{\mathbf{u}}(g_L \circ \cdots \circ g_{j+1}(\cdot)) \prod_{i=j+1}^{L} |\det(\mathbf{J}_{g_i})|$, where $j \in \{0, \cdots, L-1\}$. It follows that:*

$$\mathbb{D}_{\mathrm{F}}\left[p_{\mathbf{x}_j} \| p_j\right] = 0 \Leftrightarrow \mathbb{D}_{\mathrm{F}}\left[p_{\mathbf{x}} \| p_0\right] = 0, \forall j \in \{1, \cdots, L-1\}. \tag{10}$$

The derivation is presented in Appendix A.1.3. In Section 5.3, we validate the effectiveness of the MaP technique on the score-matching methods formulated in Eqs. (5)-(7) through an ablation analysis. Please note that MaP does not affect maximum likelihood training, since it always satisfies $\mathbb{D}_{\mathrm{KL}}\left[p_{\mathbf{x}_j} \| p_j\right] = \mathbb{D}_{\mathrm{KL}}\left[p_{\mathbf{x}} \| p_0\right], \forall j \in \{1, \cdots, L-1\}$ as revealed in Lemma A.12.

**EMA.** In addition to the MaP technique, we have also found that the exponential moving average (EMA) technique introduced in [36] is effective in improving the training stability. EMA enhances the stability through smoothly updating the parameters based on $\tilde{\theta} \leftarrow m\tilde{\theta} + (1-m)\theta_i$ at each training iteration, where $\tilde{\theta}$ is a set of shadow parameters [36], $\theta_i$ is the model's parameters at iteration $i$, and $m$ is the momentum parameter. In our experiments presented in Section 5, we adopt $m = 0.999$ for both EBFlow and the baselines.

# 5   Experiments

In the following experiments, we first compare the training efficiency of the baselines trained with $\mathcal{L}_{\mathrm{ML}}$ and EBFlow trained with $\mathcal{L}_{\mathrm{SML}}$, $\mathcal{L}_{\mathrm{SSM}}$, $\mathcal{L}_{\mathrm{FDSSM}}$, and $\mathcal{L}_{\mathrm{DSM}}$ to validate the effectiveness of the proposed method in Sections 5.1 and 5.2. Then, in Section 5.3, we provide an ablation analysis of the techniques introduced in Section 4.2, and a performance comparison between EBFlow and a number of related studies [7, 16, 17]. Finally, in Section 5.4, we discuss how EBFlow can be applied to generation tasks. Please note that the performance comparison with [8–12, 29] is omitted, since their methods only support specialized linear layers and are not applicable to the employed model architecture [7] that involves fully-connected layers. The differences between EBFlow, the baseline, and the related studies are summarized in Table A4 in the appendix. The sampling process involved in the calculation of $\mathcal{L}_{\mathrm{SML}}$ is implemented by $g^{-1}(\mathbf{u}\,;\theta)$, where $\mathbf{u} \sim p_{\mathbf{u}}$. The transformation $g(\cdot\,;\theta)$ for each task is designed such that $\mathcal{S}_l \neq \phi$ and $\mathcal{S}_n \neq \phi$. For more details about the experimental setups, please refer to Appendix A.2.

## 5.1   Density Estimation on Two-Dimensional Synthetic Examples

In this experiment, we examine the performance of EBFlow and its baseline on three two-dimensional synthetic datasets. These data distributions are formed using Gaussian smoothing kernels to ensure $p_{\mathbf{x}}(\mathbf{x})$ is continuous and the true score function $\frac{\partial}{\partial \mathbf{x}} \log p_{\mathbf{x}}(\mathbf{x})$ is well defined. The model $g(\cdot\,;\theta)$ is constructed using the Glow model architecture [4], which consists of actnorm layers, affine coupling layers, and fully-connected layers. The performance are evaluated in terms of the KL divergence and the Fisher divergence between $p_{\mathbf{x}}(\mathbf{x})$ and $p(\mathbf{x};\theta)$ using independent and identically distributed (i.i.d.) testing sample points.

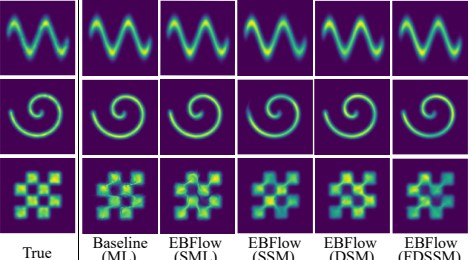

Figure 1: The visualized density functions on the Sine, Swirl, and Checkerboard datasets. The column 'True' illustrates the visualization of the true density functions.

Table 1 and Fig. 1 demonstrate the results of the above setting. The results show that the performance of EBFlow trained with $\mathcal{L}_{\mathrm{SSM}}$, $\mathcal{L}_{\mathrm{FDSSM}}$, and $\mathcal{L}_{\mathrm{DSM}}$ in terms of KL divergence is on par with those trained using $\mathcal{L}_{\mathrm{SML}}$ as well as the baselines trained using $\mathcal{L}_{\mathrm{ML}}$. These results validate the efficacy of training EBFlow with score matching.

Table 1: The evaluation results in terms of KL-divergence and Fisher-divergence of the flow-based models trained with $\mathcal{L}_{\text{ML}}$, $\mathcal{L}_{\text{SML}}$, $\mathcal{L}_{\text{SSM}}$, $\mathcal{L}_{\text{DSM}}$, and $\mathcal{L}_{\text{FDSSM}}$ on the Sine, Swirl, and Checkerboard datasets. The results are reported as the mean and 95% confidence interval of three independent runs.

| Dataset | Metric | Baseline (ML) | EBFlow (SML) | EBFlow (SSM) | EBFlow (DSM) | EBFlow (FDSSM) |
|---|---|---|---|---|---|---|
| Sine | Fisher Divergence ($\downarrow$) | 6.86 ± 0.73 e-1 | 6.65 ± 1.05 e-1 | **6.25 ± 0.84 e-1** | 6.66 ± 0.44 e-1 | 6.66 ± 1.33 e-1 |
| | KL Divergence ($\downarrow$) | **4.56 ± 0.00 e+0** | **4.56 ± 0.00 e+0** | **4.56 ± 0.01 e+0** | 4.57 ± 0.02 e+0 | 4.57 ± 0.01 e+0 |
| Swirl | Fisher Divergence ($\downarrow$) | 1.42 ± 0.48 e+0 | 1.42 ± 0.53 e+0 | 1.35 ± 0.10 e+0 | **1.34 ± 0.06 e+0** | 1.37 ± 0.07 e+0 |
| | KL Divergence ($\downarrow$) | **4.21 ± 0.00 e+0** | **4.21 ± 0.01 e+0** | 4.25 ± 0.04 e+0 | 4.22 ± 0.02 e+0 | 4.25 ± 0.08 e+0 |
| Checkerboard | Fisher Divergence ($\downarrow$) | 7.24 ± 11.50 e+1 | 1.23 ± 0.75 e+0 | 7.07 ± 1.93 e-1 | **7.03 ± 1.99 e-1** | 7.08 ± 1.62 e-1 |
| | KL Divergence ($\downarrow$) | **4.80 ± 0.02 e+0** | 4.81 ± 0.02 e+0 | 4.85 ± 0.05 e+0 | 4.82 ± 0.05 e+0 | 4.83 ± 0.03 e+0 |

Table 2: The evaluation results in terms of the performance (i.e., NLL and Bits/Dim) and the throughput (i.e., Batch/Sec.) of the FC-based and CNN-based models trained with the baseline and the proposed method on MNIST and CIFAR-10. Each result is reported in terms of the mean and 95% confidence interval of three independent runs after $\theta$ is converged. The throughput is measured on NVIDIA Tesla V100 GPUs.

| MNIST ($D = 784$) | | | | | | | | | | |
|---|---|---|---|---|---|---|---|---|---|---|
| Model | FC-based | | | | | CNN-based | | | | |
| Num. Param. | 1.230 M | | | | | 0.027 M | | | | |
| Method | Baseline (ML) | EBFlow (SML) | EBFlow (SSM) | EBFlow (DSM) | EBFlow (FDSSM) | Baseline (ML) | EBFlow (SML) | EBFlow (SSM) | EBFlow (DSM) | EBFlow (FDSSM) |
| NLL ($\downarrow$) | 1092.4 ± 0.1 | **1092.3 ± 0.6** | 1092.8 ± 0.3 | 1099.2 ± 0.2 | 1104.1 ± 0.5 | 1101.3 ± 1.3 | **1098.3 ± 6.6** | 1107.5 ± 1.4 | 1109.5 ± 2.4 | 1122.1 ± 3.1 |
| Bits/Dim ($\downarrow$) | **2.01 ± 0.00** | **2.01 ± 0.00** | **2.01 ± 0.00** | 2.02 ± 0.00 | 2.03 ± 0.00 | 2.03 ± 0.00 | **2.02 ± 0.01** | 2.03 ± 0.00 | 2.04 ± 0.00 | 2.06 ± 0.01 |
| Batch/Sec. ($\uparrow$) | 8.00 | 12.27 | 33.11 | 66.67 | **130.21** | 0.21 | 0.29 | 7.09 | 18.32 | **38.76** |
| CIFAR-10 ($D = 3,072$) | | | | | | | | | | |
| Model | FC-based | | | | | CNN-based | | | | |
| Num. Param. | 18.881 M | | | | | 0.241 M | | | | |
| Method | Baseline (ML) | EBFlow (SML) | EBFlow (SSM) | EBFlow (DSM) | EBFlow (FDSSM) | Baseline (ML) | EBFlow (SML) | EBFlow (SSM) | EBFlow (DSM) | EBFlow (FDSSM) |
| NLL ($\downarrow$) | **11912.9 ± 10.5** | 11915.6 ± 5.6 | 11917.7 ± 15.5 | 11940.0 ± 6.6 | 12347.8 ± 6.8 | **11408.7 ± 26.7** | 11553.6 ± 151.7 | 11435.5 ± 12.0 | 11462.3 ± 7.9 | 11766.0 ± 36.8 |
| Bits/Dim ($\downarrow$) | **5.59 ± 0.00** | 5.60 ± 0.00 | 5.60 ± 0.01 | 5.61 ± 0.00 | 5.80 ± 0.00 | **5.36 ± 0.01** | 5.41 ± 0.07 | 5.37 ± 0.00 | 5.38 ± 0.00 | 5.54 ± 0.02 |
| Batch/Sec. ($\uparrow$) | 5.05 | 7.35 | 29.85 | 57.14 | **62.50** | 0.02 | 0.03 | 7.35 | 18.41 | **39.84** |

## 5.2 Efficiency Evaluation on the MNIST and CIFAR-10 Datasets

In this section, we inspect the influence of data dimension $D$ on the training efficiency of flow-based models. To provide a thorough comparison, we employ two types of model architectures and train them on two datasets with different data dimensions: the MNIST [19] ($D = 1 \times 28 \times 28$) and CIFAR-10 [37] ($D = 3 \times 32 \times 32$) datasets.

The first model architecture is exactly the same as that adopted by [7]. It is an architecture consisting of two fully-connected layers and a smoothed leaky ReLU non-linear layer in between. The second model is a parametrically efficient variant of the first model. It replaces the fully-connected layers with convolutional layers and increases the depth of the model to six convolutional blocks. Between every two convolutional blocks, a squeeze operation [2] is inserted to enlarge the receptive field. In the following paragraphs, we refer to these models as 'FC-based' and 'CNN-based' models, respectively.

The performance of the FC-based and CNN-based models is measured using the negative log likelihood (NLL) metric (i.e., $\mathbb{E}_{p_{\mathsf{x}}(\boldsymbol{x})}[-\log p(\boldsymbol{x};\theta)]$), which differs from the intractable KL divergence by a constant. In addition, its normalized variant, the Bits/Dim metric [38], is also measured and reported. The algorithms are implemented using `PyTorch` [39] with automatic differentiation [40], and the runtime is measured on NVIDIA Tesla V100 GPUs. In the subsequent paragraphs, we assess the models through scalability analysis, performance evaluation, and training efficiency examination.

**Scalability.** To demonstrate the scalability of KL-divergence-based (i.e., $\mathcal{L}_{\text{ML}}$ and $\mathcal{L}_{\text{SML}}$) and Fisher-divergence-based (i.e., $\mathcal{L}_{\text{SSM}}$, $\mathcal{L}_{\text{DSM}}$, and $\mathcal{L}_{\text{FDSSM}}$) objectives used in EBFlow and the baseline method, we first present a runtime comparison for different choices of the input data size $D$. The results presented in Fig. 2 (a) reveal that Fisher-divergence-based objectives can be computed more efficiently than KL-divergence-based objectives. Moreover, the sampling-based objective $\mathcal{L}_{\text{SML}}$ used in EBFlow, which excludes the calculation of $Z(\theta)$ in the computational graph, can be computed slightly faster than $\mathcal{L}_{\text{ML}}$ adopted by the baseline.

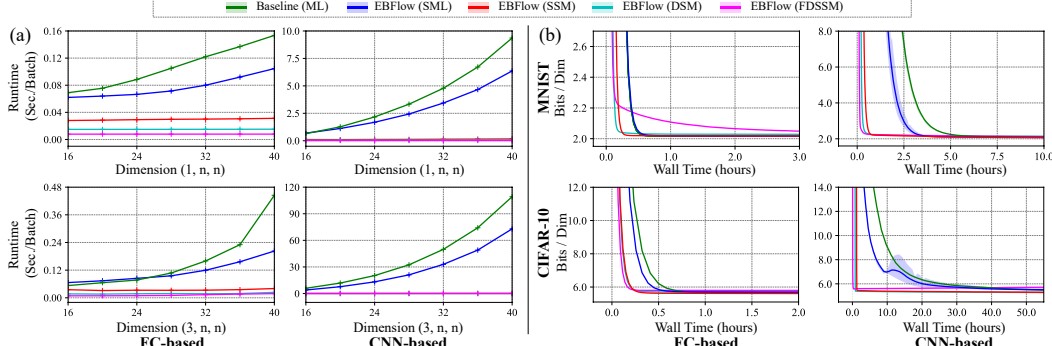

Figure 2: (a) A runtime comparison of calculating the gradients of different objectives for different input sizes ($D$). The input sizes are $(1, n, n)$ and $(3, n, n)$, with the x-axis in the figures representing $n$. In the format $(c, h, w)$, the first value indicates the number of channels, while the remaining values correspond to the height and width of the input data. The curves depict the evaluation results in terms of the mean of three independent runs. (b) A comparison of the training efficiency of the FC-based and CNN-based models evaluated on the validation set of MNIST and CIFAR-10. Each curve and the corresponding shaded area depict the mean and confidence interval of three independent runs.

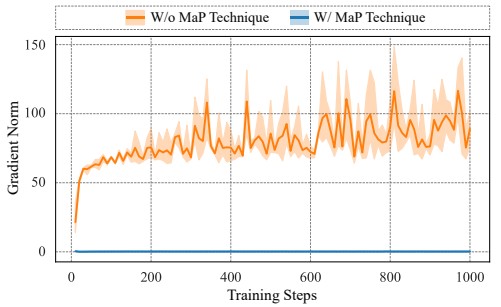

Figure 3: The norm of $\frac{\partial}{\partial \theta} \mathcal{L}_{\text{SSM}}(\theta)$ of an FC-based model trained on the MNIST dataset. The curves and shaded area depict the mean and 95% confidence interval of three independent runs.

Table 3: The results in terms of NLL of the FC-based and CNN-based models trained using SSM, DSM, and FDSSM losses on MNIST. The performance is reported in terms of the means and 95% confidence intervals of three independent runs.

| FC-based | | | | |
|---|---|---|---|---|
| EMA | MaP | EBFlow(SSM) | EBFlow(DSM) | EBFlow(FDSSM) |
| | | $1757.5 \pm 28.0$ | $4660.3 \pm 19.8$ | $3267.0 \pm 99.2$ |
| ✓ | | $1720.5 \pm 0.8$ | $4455.0 \pm 1.6$ | $3166.3 \pm 17.3$ |
| ✓ | ✓ | $\mathbf{1092.8 \pm 0.3}$ | $\mathbf{1099.2 \pm 0.2}$ | $\mathbf{1104.1 \pm 0.5}$ |

| CNN-based | | | | |
|---|---|---|---|---|
| EMA | MaP | EBFlow(SSM) | EBFlow(DSM) | EBFlow(FDSSM) |
| | | $3518.0 \pm 33.9$ | $3170.0 \pm 7.2$ | $3593.3 \pm 12.5$ |
| ✓ | | $3504.5 \pm 2.4$ | $3180.0 \pm 2.9$ | $3560.3 \pm 1.7$ |
| ✓ | ✓ | $\mathbf{1107.5 \pm 1.4}$ | $\mathbf{1109.5 \pm 2.6}$ | $\mathbf{1122.1 \pm 3.1}$ |

**Performance.** Table 2 demonstrates the performance of the FC-based and CNN-based models in terms of NLL on the MNIST and CIFAR-10 datasets. The results show that the models trained with Fisher-divergence-based objectives are able to achieve similar performance as those trained with KL-divergence-based objectives. Among the Fisher-divergence-based objectives, the models trained using $\mathcal{L}_{\text{SSM}}$ and $\mathcal{L}_{\text{DSM}}$ are able to achieve better performance in comparison to those trained using $\mathcal{L}_{\text{FDSSM}}$. The runtime and performance comparisons above suggest that $\mathcal{L}_{\text{SSM}}$ and $\mathcal{L}_{\text{DSM}}$ can deliver better training efficiency than $\mathcal{L}_{\text{ML}}$ and $\mathcal{L}_{\text{SML}}$, since the objectives can be calculated faster while maintaining the models' performance on the NLL metric.

**Training Efficiency.** Fig. 2 (b) presents the trends of NLL versus training wall time when $\mathcal{L}_{\text{ML}}$, $\mathcal{L}_{\text{SML}}$, $\mathcal{L}_{\text{SSM}}$, $\mathcal{L}_{\text{DSM}}$, and $\mathcal{L}_{\text{FDSSM}}$ are adopted as the objectives. It is observed that EBFlow trained with SSM and DSM consistently attain better NLL in the early stages of the training. The improvement is especially notable when both $D$ and $L$ are large, as revealed for the scenario of training CNN-based models on the CIFAR-10 dataset. These experimental results provide evidence to support the use of score-matching methods for optimizing EBFlow.

### 5.3 Analyses and Comparisons

**Ablation Study.** Table 3 presents the ablation results that demonstrate the effectiveness of the EMA and MaP techniques. It is observed that EMA is effective in reducing the variances. In addition, MaP significantly improves the overall performance. To further illustrate the influence of the proposed MaP technique on the score-matching methods, we compare the optimization pro-

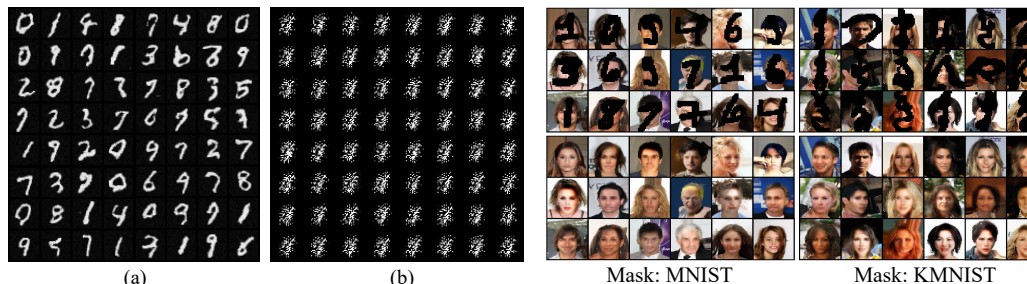

| (a) | (b) | Mask: MNIST | Mask: KMNIST |

Figure 4: A qualitative comparison between (a) our model (NLL=728) and (b) the model in [17] (NLL=1,637) on the inverse generation task.

Figure 5: A qualitative demonstration of the FC-based model trained using $\mathcal{L}_{\text{DSM}}$ on the data imputation task.

cesses with $\frac{\partial}{\partial\theta}\mathbb{D}_{\text{F}}\left[p_{\mathbf{x}_k}\|p_k\right]$ and $\frac{\partial}{\partial\theta}\mathbb{D}_{\text{F}}\left[p_{\mathbf{x}}\|p_0\right] = \frac{\partial}{\partial\theta}\mathbb{E}_{p_{\mathbf{x}_k}(\boldsymbol{x}_k)}[\frac{1}{2}\|(\frac{\partial}{\partial\boldsymbol{x}_k}\log(\frac{p_{\mathbf{x}_k}(\boldsymbol{x}_k)}{p_k(\boldsymbol{x}_k)}))\prod_{i=1}^{k}\mathbf{J}_{g_i}\|^2]$ (i.e., Lemma A.13) by depicting the norm of their unbiased estimators $\frac{\partial}{\partial\theta}\mathcal{L}_{\text{SSM}}(\theta)$ calculated with and without applying the MaP technique in Fig. 3. It is observed that the magnitude of $\left\|\frac{\partial}{\partial\theta}\mathcal{L}_{\text{SSM}}(\theta)\right\|$ significantly decreases when MaP is incorporated into the training process. This could be attributed to the fact that the calculation of $\frac{\partial}{\partial\theta}\mathbb{D}_{\text{F}}\left[p_{\mathbf{x}_k}\|p_k\right]$ excludes the calculation of $\prod_{i=1}^{k}\mathbf{J}_{g_i}$ in $\frac{\partial}{\partial\theta}\mathbb{D}_{\text{F}}\left[p_{\mathbf{x}}\|p_0\right]$, which involves computing the derivatives of the numerically sensitive logit pre-processing layer.

**Comparison with Related Works.** Table 4 compares the performance of our method with a number of related works on the MNIST dataset. Our models trained with score-matching objectives using the same model architecture exhibit improved performance in comparison to the relative gradient method [7]. In addition, when compared to the results in [16] and [17], our models deliver significantly improved performance over them. Please note that the results of [7, 16, 17] presented in Table 4 are obtained from their original papers.

Table 4: A comparison of performance and training complexity between EBFlow and a number of related works [16, 7, 17] on the MNIST dataset.

|  | Method | Complexity | NLL ($\downarrow$) |
|---|---|---|---|
| $\mathbb{D}_{\text{KL}}$-Based | Baseline (ML) | $\mathcal{O}(D^3 L)$ | $1092.4 \pm 0.1$ |
|  | EBFlow (SML) | $\mathcal{O}(D^3 L)$ | $\mathbf{1092.3 \pm 0.6}$ |
|  | Relative Grad. [7] | $\mathcal{O}(D^2 L)$ | $1375.2 \pm 1.4$ |
| $\mathbb{D}_{\text{F}}$-Based | EBFlow (SSM) | $\mathcal{O}(D^2 L)$ | $1092.8 \pm 0.3$ |
|  | EBFlow (DSM) | $\mathcal{O}(D^2 L)$ | $1099.2 \pm 0.2$ |
|  | EBFlow (FDSSM) | $\mathcal{O}(D^2 L)$ | $1104.1 \pm 0.5$ |
|  | SSM [16] | - | $3355$ |
|  | DSM [17] | - | $3398 \pm 1343$ |
|  | FDSSM [17] | - | $1647 \pm 306$ |

## 5.4 Application to Generation Tasks

The sampling process of EBFlow can be accomplished through the inverse function or an MCMC process. The former is a typical generation method adopted by flow-based models, while the latter is a more flexible sampling process that allows conditional generation without re-training the model. In the following paragraphs, we provide detailed explanations and visualized results of these tasks.

**Inverse Generation.** One benefit of flow-based models is that $g^{-1}$ can be directly adopted as a generator. While inverting the weight matrices in linear transformations typically demands time complexity of $\mathcal{O}(D^3 L)$, these inverse matrices are only required to be computed once $\theta$ has converged, and can then be reused for subsequent inferences In this experiment, we adopt the Glow [4] model architecture and train it using our method with $\mathcal{L}_{\text{SSM}}$ on the MNIST dataset. We compare our visualized results with the current best flow-based model trained using the score matching objective [17]. The results of [17] are generated using their officially released code with their best setup (i.e., FDSSM). As presented in Fig. 4, the results generated using our model demonstrate significantly better visual quality than those of [17].

**MCMC Generation.** In comparison to the inverse generation method, the MCMC sampling process is more suitable for conditional generation tasks such as data imputation due to its flexibility [41]. For the imputation task, a data vector $\boldsymbol{x}$ is separated as an observable part $\boldsymbol{x}_O$ and a masked part $\boldsymbol{x}_M$. The goal of imputation is to generate the masked part $\boldsymbol{x}_M$ based on the observable part $\boldsymbol{x}_O$. To achieve this goal, one can perform a Langevin MCMC process to update $\boldsymbol{x}_M$ according to the gradient of the energy function $\frac{\partial}{\partial\boldsymbol{x}}E(\boldsymbol{x};\theta)$. Given a noise vector $\boldsymbol{z}$ sampled from $\mathcal{N}(\mathbf{0}, \boldsymbol{I})$ and a

small step size $\alpha$, the process iteratively updates $\boldsymbol{x}_M$ based on the following equation:

$$\boldsymbol{x}_M^{(t+1)} = \boldsymbol{x}_M^{(t)} - \alpha \frac{\partial}{\partial \boldsymbol{x}_M^{(t)}} E(\boldsymbol{x}_O, \boldsymbol{x}_M^{(t)}; \theta) + \sqrt{2\alpha} \boldsymbol{z}, \qquad (11)$$

where $\boldsymbol{x}_M^{(t)}$ represents $\boldsymbol{x}_M$ at iteration $t \in \{1, \cdots, T\}$, and $T$ is the total number of iterations. MCMC generation requires an overall cost of $\mathcal{O}(TD^2L)$, potentially more economical than the $\mathcal{O}(D^3L)$ computation of the inverse generation method. Fig. 5 depicts the imputation results of the FC-based model trained using $\mathcal{L}_{\mathrm{DSM}}$ on the CelebA [42] dataset ($D = 3 \times 64 \times 64$). In this example, we implement the masking part $\boldsymbol{x}_M$ using the data from the KMNIST [43] and MNIST [19] datasets.

## 6 Conclusion

In this paper, we presented EBFlow, a new flow-based modeling approach that associates the parameterization of flow-based and energy-based models. We showed that by optimizing EBFlow with score-matching objectives, the computation of Jacobian determinants for linear transformations can be bypassed, resulting in an improved training time complexity. In addition, we demonstrated that the training stability and performance can be effectively enhanced through the MaP and EMA techniques. Based on the improvements in both theoretical time complexity and empirical performance, our method exhibits superior training efficiency compared to maximum likelihood training.

## Acknowledgement

The authors gratefully acknowledge the support from the National Science and Technology Council (NSTC) in Taiwan under grant number MOST 111-2223-E-007-004-MY3, as well as the financial support from MediaTek Inc., Taiwan. The authors would also like to express their appreciation for the donation of the GPUs from NVIDIA Corporation and NVIDIA AI Technology Center (NVAITC) used in this work. Furthermore, the authors extend their gratitude to the National Center for High-Performance Computing (NCHC) for providing the necessary computational and storage resources.

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

# A Appendix

## A.1 Derivations

In the following subsections, we provide theoretical derivations. In Section A.1.1, we discuss the asymptotic convergence properties as well as the assumptions of score-matching methods. In Section A.1.2, we elaborate on the formulation of EBFlow (i.e., Eqs. (8) and (9)), and provide a explanation of their interpretation. Finally, in Section A.1.3, we present a theoretical analysis of KL divergence and Fisher divergence, and discuss the underlying mechanism behind the proposed MaP technique.

### A.1.1 Asymptotic Convergence Property of Score Matching

In this subsection, we provide a formal description of the *consistency* property of score matching. The description follows [16] and the notations are replaced with those used in this paper. The regularity conditions for $p(\cdot\,;\theta)$ are defined in Assumptions A.1~A.7. In the following paragraph, the parameter space is defined as $\Theta$. In addition, $s(\boldsymbol{x};\theta) \triangleq \frac{\partial}{\partial \boldsymbol{x}} \log p(\boldsymbol{x};\theta) = -\frac{\partial}{\partial \boldsymbol{x}} E(\boldsymbol{x};\theta)$ represents the score function. $\hat{\mathcal{L}}_{\mathrm{SM}}(\theta) \triangleq \frac{1}{N} \sum_{k=1}^{N} f(\mathbf{x}_k;\theta)$ denotes an unbiased estimator of $\mathcal{L}_{\mathrm{SM}}(\theta)$, where $f(\boldsymbol{x};\theta) \triangleq \frac{1}{2} \left\| \frac{\partial}{\partial \boldsymbol{x}} E(\boldsymbol{x};\theta) \right\|^2 - \mathrm{Tr}\left( \frac{\partial^2}{\partial \boldsymbol{x}^2} E(\boldsymbol{x};\theta) \right) = \frac{1}{2} \left\| s(\boldsymbol{x};\theta) \right\|^2 + \mathrm{Tr}\left( \frac{\partial}{\partial \boldsymbol{x}} s(\boldsymbol{x};\theta) \right)$ and $\{\mathbf{x}_1, \cdots, \mathbf{x}_N\}$ represents a collection of i.i.d. samples drawn from $p_{\mathbf{x}}$. For notational simplicity, we denote $\partial h(\boldsymbol{x};\theta) \triangleq \frac{\partial}{\partial \boldsymbol{x}} h(\boldsymbol{x};\theta)$ and $\partial_i h_j(\boldsymbol{x};\theta) \triangleq \frac{\partial}{\partial \boldsymbol{x}_i} h_j(\boldsymbol{x};\theta)$, where $h_j(\boldsymbol{x};\theta)$ denotes the $j$-th element of $h$.

**Assumption A.1.** (Positiveness) $p(\boldsymbol{x};\theta) > 0$ and $p_{\mathbf{x}}(\boldsymbol{x}) > 0, \forall \theta \in \Theta, \forall \boldsymbol{x} \in \mathbb{R}^D$.

**Assumption A.2.** (Regularity of the score functions) The parameterized score function $s(\boldsymbol{x};\theta)$ and the true score function $\frac{\partial}{\partial \boldsymbol{x}} \log p_{\mathbf{x}}(\boldsymbol{x})$ are both continuous and differentiable. In addition, their expectations $\mathbb{E}_{p_{\mathbf{x}}(\boldsymbol{x})} [s(\boldsymbol{x};\theta)]$ and $\mathbb{E}_{p_{\mathbf{x}}(\boldsymbol{x})} \left[ \frac{\partial}{\partial \boldsymbol{x}} \log p_{\mathbf{x}}(\boldsymbol{x}) \right]$ are finite. (i.e., $\mathbb{E}_{p_{\mathbf{x}}(\boldsymbol{x})} [s(\boldsymbol{x};\theta)] < \infty$ and $\mathbb{E}_{p_{\mathbf{x}}(\boldsymbol{x})} \left[ \frac{\partial}{\partial \boldsymbol{x}} \log p_{\mathbf{x}}(\boldsymbol{x}) \right] < \infty$)

**Assumption A.3.** (Boundary condition) $\lim_{\|\boldsymbol{x}\| \to \infty} p_{\mathbf{x}}(\boldsymbol{x}) s(\boldsymbol{x};\theta) = 0, \forall \theta \in \Theta$.

**Assumption A.4.** (Compactness) The parameter space $\Theta$ is compact.

**Assumption A.5.** (Identifiability) There exists a set of parameters $\theta^*$ such that $p_{\mathbf{x}}(\boldsymbol{x}) = p(\boldsymbol{x};\theta^*)$, where $\theta^* \in \Theta, \forall \boldsymbol{x} \in \mathbb{R}^D$.

**Assumption A.6.** (Uniqueness) $\theta \neq \theta^* \Leftrightarrow p(\boldsymbol{x};\theta) \neq p(\boldsymbol{x};\theta^*)$, where $\theta, \theta^* \in \Theta, \boldsymbol{x} \in \mathbb{R}^D$.

**Assumption A.7.** (Lipschitzness of $f$) The function $f$ is Lipschitz continuous w.r.t. $\theta$, i.e., $|f(\boldsymbol{x};\theta_1) - f(\boldsymbol{x};\theta_2)| \leq L(\boldsymbol{x}) \|\theta_1 - \theta_2\|_2, \forall \theta_1, \theta_2 \in \Theta$, where $L(\boldsymbol{x})$ represents a Lipschitz constant satisfying $\mathbb{E}_{p_{\mathbf{x}}(\boldsymbol{x})} [L(\boldsymbol{x})] < \infty$.

**Theorem A.8.** *(Consistency of a score-matching estimator [16]) The score-matching estimator $\theta_N \triangleq \operatorname{argmin}_{\theta \in \Theta} \hat{\mathcal{L}}_{\mathrm{SM}}$ is consistent, i.e.,*

$$\theta_N \xrightarrow{p} \theta^*, \text{ as } N \to \infty.$$

Assumptions A.1~A.3 are the conditions that ensure $\frac{\partial}{\partial \theta} \mathbb{D}_{\mathrm{F}} [p_{\mathbf{x}}(\boldsymbol{x}) \| p(\boldsymbol{x};\theta)] = \frac{\partial}{\partial \theta} \mathcal{L}_{\mathrm{SM}}(\theta)$. Assumptions A.4~A.7 lead to the uniform convergence property [16] of a score-matching estimator, which gives rise to the *consistency* property. The detailed derivation can be found in Corollary 1 in [16]. In the following Lemma A.9 and Proposition A.10, we examine the sufficient condition for $g$ and $p_{\mathbf{u}}$ to satisfy Assumption A.7.

**Lemma A.9.** *(Sufficient condition for the Lipschitzness of $f$) The function $f(\boldsymbol{x};\theta) = \frac{1}{2} \|s(\boldsymbol{x};\theta)\|^2 + \mathrm{Tr}\left( \frac{\partial}{\partial \boldsymbol{x}} s(\boldsymbol{x};\theta) \right)$ is Lipschitz continuous if the score function $s(\boldsymbol{x};\theta)$ satisfies the following conditions: $\forall \theta, \theta_1, \theta_2 \in \Theta, \forall i \in \{1, \cdots, D\}$,*

$$\|s(\boldsymbol{x};\theta)\|_2 \leq L_1(\boldsymbol{x}),$$
$$\|s(\boldsymbol{x};\theta_1) - s(\boldsymbol{x};\theta_2)\|_2 \leq L_2(\boldsymbol{x}) \|\theta_1 - \theta_2\|_2,$$
$$\|\partial_i s(\boldsymbol{x};\theta_1) - \partial_i s(\boldsymbol{x};\theta_2)\|_2 \leq L_3(\boldsymbol{x}) \|\theta_1 - \theta_2\|_2,$$

*where $L_1, L_2$, and $L_3$ are Lipschitz constants satisfying $\mathbb{E}_{p_{\mathbf{x}}(\boldsymbol{x})} [L_1(\boldsymbol{x})] < \infty$, $\mathbb{E}_{p_{\mathbf{x}}(\boldsymbol{x})} [L_2(\boldsymbol{x})] < \infty$, and $\mathbb{E}_{p_{\mathbf{x}}(\boldsymbol{x})} [L_3(\boldsymbol{x})] < \infty$.*

*Proof.* The Lipschitzness of $f$ can be guaranteed by ensuring the Lipschitzness of $\|s(\boldsymbol{x};\theta)\|_2^2$ and $\mathrm{Tr}\left(\partial s(\boldsymbol{x};\theta)\right)$.

**Step 1.** (Lipschitzness of $\|s(\boldsymbol{x};\theta)\|_2^2$)

$$
\left| \|s(\boldsymbol{x};\theta_1)\|_2^2 - \|s(\boldsymbol{x};\theta_2)\|_2^2 \right|
$$

$$
= \left| s(\boldsymbol{x};\theta_1)^T s(\boldsymbol{x};\theta_1) - s(\boldsymbol{x};\theta_2)^T s(\boldsymbol{x};\theta_2) \right|
$$

$$
= \left| \left( s(\boldsymbol{x};\theta_1)^T s(\boldsymbol{x};\theta_1) - s(\boldsymbol{x};\theta_1)^T s(\boldsymbol{x};\theta_2) \right) + \left( s(\boldsymbol{x};\theta_1)^T s(\boldsymbol{x};\theta_2) - s(\boldsymbol{x};\theta_2)^T s(\boldsymbol{x};\theta_2) \right) \right|
$$

$$
= \left| s(\boldsymbol{x};\theta_1)^T \left( s(\boldsymbol{x};\theta_1) - s(\boldsymbol{x};\theta_2) \right) + s(\boldsymbol{x};\theta_2)^T \left( s(\boldsymbol{x};\theta_1) - s(\boldsymbol{x};\theta_2) \right) \right|
$$

$$
\overset{(i)}{\leq} \left| s(\boldsymbol{x};\theta_1)^T \left( s(\boldsymbol{x};\theta_1) - s(\boldsymbol{x};\theta_2) \right) \right| + \left| s(\boldsymbol{x};\theta_2)^T \left( s(\boldsymbol{x};\theta_1) - s(\boldsymbol{x};\theta_2) \right) \right|
$$

$$
\overset{(ii)}{\leq} \|s(\boldsymbol{x};\theta_1)\|_2 \|s(\boldsymbol{x};\theta_1) - s(\boldsymbol{x};\theta_2)\|_2 + \|s(\boldsymbol{x};\theta_2)\|_2 \|s(\boldsymbol{x};\theta_1) - s(\boldsymbol{x};\theta_2)\|_2
$$

$$
\overset{(iii)}{\leq} L_1(\boldsymbol{x}) \|s(\boldsymbol{x};\theta_1) - s(\boldsymbol{x};\theta_2)\|_2 + L_1(\boldsymbol{x}) \|s(\boldsymbol{x};\theta_1) - s(\boldsymbol{x};\theta_2)\|_2
$$

$$
\overset{(iii)}{\leq} 2L_1(\boldsymbol{x})L_2(\boldsymbol{x}) \|\theta_1 - \theta_2\|_2 ,
$$

where $(i)$ is based on triangle inequality, $(ii)$ is due to Cauchy–Schwarz inequality, and $(iii)$ follows from the listed assumptions.

**Step 2.** (Lipschitzness of $\mathrm{Tr}\left(\partial s(\boldsymbol{x};\theta)\right)$)

$$
|\mathrm{Tr}\left(\partial s(\boldsymbol{x};\theta_1)\right) - \mathrm{Tr}\left(\partial s(\boldsymbol{x};\theta_2)\right)| = |\mathrm{Tr}\left(\partial s(\boldsymbol{x};\theta_1) - \partial s(\boldsymbol{x};\theta_2)\right)|
$$

$$
\overset{(i)}{\leq} D \|\partial s(\boldsymbol{x};\theta_1) - \partial s(\boldsymbol{x};\theta_2)\|_2
$$

$$
\overset{(ii)}{\leq} D \sqrt{\sum_i \|\partial_i s(\boldsymbol{x};\theta_1) - \partial_i s(\theta_2)\|_2^2}
$$

$$
\overset{(iii)}{\leq} D \sqrt{D L_3^2(\boldsymbol{x}) \|\theta_1 - \theta_2\|_2^2}
$$

$$
= D\sqrt{D} L_3(\boldsymbol{x}) \|\theta_1 - \theta_2\|_2
$$

where $(i)$ holds by Von Neumann's trace inequality. $(ii)$ is due to the property $\|A\|_2 \leq \sqrt{\sum_i \|\boldsymbol{a}_i\|_2^2}$, where $\boldsymbol{a}_i$ is the column vector of $A$. $(iii)$ holds by the listed assumptions.

Based on Steps 1 and 2, the Lipschitzness of $f$ is guaranteed, since

$$
|f(\boldsymbol{x};\theta_1) - f(\boldsymbol{x};\theta_2)| = \left| \frac{1}{2} \|s(\boldsymbol{x};\theta_1)\|^2 + \mathrm{Tr}\left( \frac{\partial}{\partial \boldsymbol{x}} s(\boldsymbol{x};\theta_1) \right) - \frac{1}{2} \|s(\boldsymbol{x};\theta_2)\|^2 - \mathrm{Tr}\left( \frac{\partial}{\partial \boldsymbol{x}} s(\boldsymbol{x};\theta_2) \right) \right|
$$

$$
= \left| \frac{1}{2} \|s(\boldsymbol{x};\theta_1)\|^2 - \frac{1}{2} \|s(\boldsymbol{x};\theta_2)\|^2 + \mathrm{Tr}\left( \frac{\partial}{\partial \boldsymbol{x}} s(\boldsymbol{x};\theta_1) \right) - \mathrm{Tr}\left( \frac{\partial}{\partial \boldsymbol{x}} s(\boldsymbol{x};\theta_2) \right) \right|
$$

$$
\leq \frac{1}{2} \left| \|s(\boldsymbol{x};\theta_1)\|^2 - \|s(\boldsymbol{x};\theta_2)\|^2 \right| + \left| \mathrm{Tr}\left( \frac{\partial}{\partial \boldsymbol{x}} s(\boldsymbol{x};\theta_1) \right) - \mathrm{Tr}\left( \frac{\partial}{\partial \boldsymbol{x}} s(\boldsymbol{x};\theta_2) \right) \right|
$$

$$
\leq L_1(\boldsymbol{x})L_2(\boldsymbol{x}) \|\theta_1 - \theta_2\|_2 + D\sqrt{D} L_3(\boldsymbol{x}) \|\theta_1 - \theta_2\|_2
$$

$$
= \left( L_1(\boldsymbol{x})L_2(\boldsymbol{x}) + D\sqrt{D} L_3(\boldsymbol{x}) \right) \|\theta_1 - \theta_2\|_2 .
$$

$\square$

**Proposition A.10.** *(Sufficient condition for the Lipschitzness of $f$) The function $f$ is Lipschitz continuous if $g(\boldsymbol{x};\theta)$ has bounded first, second, and third-order derivatives, i.e., $\forall i, j \in \{1, \cdots, D\}$, $\forall \theta \in \Theta$.*

$$
\|\mathbf{J}_g(\boldsymbol{x};\theta)\|_2 \leq l_1(\boldsymbol{x}), \|\partial_i \mathbf{J}_g(\boldsymbol{x};\theta)\|_2 \leq l_2(\boldsymbol{x}), \|\partial_i \partial_j \mathbf{J}_g(\boldsymbol{x};\theta)\|_2 \leq l_3(\boldsymbol{x}),
$$

*and smooth enough on $\Theta$, i.e., $\theta_1, \theta_2 \in \Theta$:*

$$
\|g(\boldsymbol{x};\theta_1) - g(\boldsymbol{x};\theta_2)\|_2 \leq r_0(\boldsymbol{x}) \|\theta_1 - \theta_2\|_2 ,
$$

$$\|\mathbf{J}_g(\boldsymbol{x};\theta_1) - \mathbf{J}_g(\boldsymbol{x};\theta_2)\|_2 \le r_1(\boldsymbol{x})\|\theta_1 - \theta_2\|_2,$$
$$\|\partial_i\mathbf{J}_g(\boldsymbol{x};\theta_1) - \partial_i\mathbf{J}_g(\boldsymbol{x};\theta_2)\|_2 \le r_2(\boldsymbol{x})\|\theta_1 - \theta_2\|_2.$$
$$\|\partial_i\partial_j\mathbf{J}_g(\boldsymbol{x};\theta_1) - \partial_i\partial_j\mathbf{J}_g(\boldsymbol{x};\theta_2)\|_2 \le r_3(\boldsymbol{x})\|\theta_1 - \theta_2\|_2.$$

*In addition, it satisfies the following conditions:*

$$\left\|\mathbf{J}_g^{-1}(\boldsymbol{x};\theta)\right\|_2 \le l_1'(\boldsymbol{x}),\ \left\|\partial_i\mathbf{J}_g^{-1}(\boldsymbol{x};\theta)\right\|_2 \le l_2'(\boldsymbol{x}),$$

$$\left\|\mathbf{J}_g^{-1}(\boldsymbol{x};\theta_1) - \mathbf{J}_g^{-1}(\boldsymbol{x};\theta_2)\right\|_2 \le r_1'(\boldsymbol{x})\|\theta_1 - \theta_2\|_2,$$

$$\left\|\partial_i\mathbf{J}_g^{-1}(\boldsymbol{x};\theta_1) - \partial_i\mathbf{J}_g^{-1}(\boldsymbol{x};\theta_2)\right\|_2 \le r_2'(\boldsymbol{x})\|\theta_1 - \theta_2\|_2,$$

*where $\mathbf{J}_g^{-1}$ represents the inverse matrix of $\mathbf{J}_g$. Furthermore, the prior distribution $p_\mathbf{u}$ satisfies:*

$$\|s_\mathbf{u}(\boldsymbol{u})\| \le t_1,\ \|\partial_i s_\mathbf{u}(\boldsymbol{u})\| \le t_2$$
$$\|s_\mathbf{u}(\boldsymbol{u}_1) - s_\mathbf{u}(\boldsymbol{u}_2)\|_2 \le t_3\|\boldsymbol{u}_1 - \boldsymbol{u}_2\|_2,$$
$$\|\partial_i s_\mathbf{u}(\boldsymbol{u}_1) - \partial_i s_\mathbf{u}(\boldsymbol{u}_2)\|_2 \le t_4\|\boldsymbol{u}_1 - \boldsymbol{u}_2\|_2,$$

*where $s_\mathbf{u}(\boldsymbol{u}) \triangleq \frac{\partial}{\partial\boldsymbol{u}}\log p_\mathbf{u}(\boldsymbol{u})$ is the score function of $p_\mathbf{u}$. The Lipschitz constants listed above (i.e., $l_1 \sim l_3$, $r_0 \sim r_3$, $l_1' \sim l_2'$, and $r_1' \sim r_2'$) have finite expectations.*

*Proof.* We show that the sufficient conditions stated in Lemma A.9 can be satisfied using the conditions listed above.

**Step 1.** (Sufficient condition of $\|s(\boldsymbol{x};\theta)\|_2 \le L_1(\boldsymbol{x})$)

Since $\|s(\boldsymbol{x};\theta)\|_2 = \left\|\frac{\partial}{\partial\boldsymbol{x}}\log p_\mathbf{u}(g(\boldsymbol{x};\theta)) + \frac{\partial}{\partial\boldsymbol{x}}\log|\det\mathbf{J}_g(\boldsymbol{x};\theta)|\right\|_2 \le \left\|\frac{\partial}{\partial\boldsymbol{x}}\log p_\mathbf{u}(g(\boldsymbol{x};\theta))\right\|_2 + \left\|\frac{\partial}{\partial\boldsymbol{x}}\log|\det\mathbf{J}_g(\boldsymbol{x};\theta)|\right\|_2$, we first demonstrate that $\left\|\frac{\partial}{\partial\boldsymbol{x}}\log p_\mathbf{u}(g(\boldsymbol{x};\theta))\right\|_2$ and $\left\|\frac{\partial}{\partial\boldsymbol{x}}\log|\det\mathbf{J}_g(\boldsymbol{x};\theta)|\right\|_2$ are both bounded.

(1.1) $\left\|\frac{\partial}{\partial\boldsymbol{x}}\log p_\mathbf{u}(g(\boldsymbol{x};\theta))\right\|_2$ is bounded:

$$\left\|\frac{\partial}{\partial\boldsymbol{x}}\log p_\mathbf{u}(g(\boldsymbol{x};\theta))\right\|_2 = \left\|(s_\mathbf{u}(g(\boldsymbol{x};\theta)))^T\mathbf{J}_g(\boldsymbol{x};\theta)\right\|_2 \le \|s_\mathbf{u}(g(\boldsymbol{x};\theta))\|_2\|\mathbf{J}_g(\boldsymbol{x};\theta)\|_2 \le t_1 l_1(\boldsymbol{x}).$$

(1.2) $\left\|\frac{\partial}{\partial\boldsymbol{x}}\log|\det\mathbf{J}_g(\boldsymbol{x};\theta)|\right\|$ is bounded:

$$\left\|\frac{\partial}{\partial\boldsymbol{x}}\log|\det\mathbf{J}_g(\boldsymbol{x};\theta)|\right\| = \left\||\det\mathbf{J}_g(\boldsymbol{x};\theta)|^{-1}\frac{\partial}{\partial\boldsymbol{x}}|\det\mathbf{J}_g(\boldsymbol{x};\theta)|\right\|$$

$$= \left\|(\det\mathbf{J}_g(\boldsymbol{x};\theta))^{-1}\frac{\partial}{\partial\boldsymbol{x}}\det\mathbf{J}_g(\boldsymbol{x};\theta)\right\|$$

$$\overset{(i)}{=} \left\|(\det\mathbf{J}_g(\boldsymbol{x};\theta))^{-1}\det\mathbf{J}_g(\boldsymbol{x};\theta)\boldsymbol{v}(\boldsymbol{x};\theta)\right\|$$

$$= \|\boldsymbol{v}(\boldsymbol{x};\theta)\|,$$

where $(i)$ is derived using Jacobi's formula, and $\boldsymbol{v}_i(\boldsymbol{x};\theta) = \mathrm{Tr}\left(\mathbf{J}_g^{-1}(\boldsymbol{x};\theta)\partial_i\mathbf{J}_g(\boldsymbol{x};\theta)\right)$.

$$\|\boldsymbol{v}(\boldsymbol{x};\theta)\| = \sqrt{\sum_i\left(\mathrm{Tr}\left(\mathbf{J}_g^{-1}(\boldsymbol{x};\theta)\partial_i\mathbf{J}_g(\boldsymbol{x};\theta)\right)\right)^2}$$

$$\overset{(i)}{\le} \sqrt{\sum_i D^2\left\|\mathbf{J}_g^{-1}(\boldsymbol{x};\theta)\partial_i\mathbf{J}_g(\boldsymbol{x};\theta)\right\|_2^2}$$

$$\overset{(ii)}{\le} \sqrt{\sum_i D^2\left\|\mathbf{J}_g^{-1}(\boldsymbol{x};\theta)\right\|_2^2\|\partial_i\mathbf{J}_g(\boldsymbol{x};\theta)\|_2^2}$$

$$\overset{(iii)}{\le} \sqrt{\sum_i D^2 l_1'^2(\boldsymbol{x})l_2^2(\boldsymbol{x})}$$

$$= \sqrt{D^3}l_1'(\boldsymbol{x})l_2(\boldsymbol{x}),$$

where $(i)$ holds by Von Neumann's trace inequality, $(ii)$ is due to the property of matrix norm, and $(iii)$ is follows from the listed assumptions.

**Step 2.** (Sufficient condition of the Lipschitzness of $s(\boldsymbol{x}; \theta)$)

Since $s(\boldsymbol{x}; \theta) = \frac{\partial}{\partial \boldsymbol{x}} \log p_{\mathbf{u}}(g(\boldsymbol{x}; \theta)) + \frac{\partial}{\partial \boldsymbol{x}} \log |\det \mathbf{J}_g(\boldsymbol{x}; \theta)|$, we demonstrate that $\frac{\partial}{\partial \boldsymbol{x}} \log p_{\mathbf{u}}(g(\boldsymbol{x}; \theta))$ and $\frac{\partial}{\partial \boldsymbol{x}} \log |\det \mathbf{J}_g(\boldsymbol{x}; \theta)|$ are both Lipschitz continuous on $\Theta$.

(2.1) Lipschitzness of $\frac{\partial}{\partial \boldsymbol{x}} \log p_{\mathbf{u}}(g(\boldsymbol{x}; \theta))$:

$$
\left\| \frac{\partial}{\partial \boldsymbol{x}} \log p_{\mathbf{u}}(g(\boldsymbol{x}; \theta_1)) - \frac{\partial}{\partial \boldsymbol{x}} \log p_{\mathbf{u}}(g(\boldsymbol{x}; \theta_2)) \right\|_2
$$

$$
= \left\| (s_{\mathbf{u}}(g(\boldsymbol{x}; \theta_1)))^T \mathbf{J}_g(\boldsymbol{x}; \theta_1) - (s_{\mathbf{u}}(g(\boldsymbol{x}; \theta_2)))^T \mathbf{J}_g(\boldsymbol{x}; \theta_2) \right\|_2
$$

$$
\overset{(i)}{\leq} \|s_{\mathbf{u}}(g(\boldsymbol{x}; \theta_1))\|_2 \|\mathbf{J}_g(\boldsymbol{x}; \theta_1) - \mathbf{J}_g(\boldsymbol{x}; \theta_2)\|_2 + \|s_{\mathbf{u}}(g(\boldsymbol{x}; \theta_1)) - s_{\mathbf{u}}(g(\boldsymbol{x}; \theta_2))\|_2 \|\mathbf{J}_g(\boldsymbol{x}; \theta_2)\|_2
$$

$$
\overset{(ii)}{\leq} t_1 r_1(\boldsymbol{x}) \|\theta_1 - \theta_2\|_2 + t_2 l_1(\boldsymbol{x}) \|g(\boldsymbol{x}; \theta_1) - g(\boldsymbol{x}; \theta_2)\|_2
$$

$$
\overset{(ii)}{\leq} t_1 r_1(\boldsymbol{x}) \|\theta_1 - \theta_2\|_2 + t_2 l_1(\boldsymbol{x}) r_0(\boldsymbol{x}) \|\theta_1 - \theta_2\|_2
$$

$$
= (t_1 r_1(\boldsymbol{x}) + t_2 l_1(\boldsymbol{x}) r_0(\boldsymbol{x})) \|\theta_1 - \theta_2\|_2,
$$

where $(i)$ is obtained using a similar derivation to Step 1 in Lemma A.9, while $(ii)$ follows from the listed assumptions.

(2.2) Lipschitzness of $\frac{\partial}{\partial \boldsymbol{x}} \log |\det \mathbf{J}_g(\boldsymbol{x}; \theta)|$:

Let $\mathbf{M}(i, \boldsymbol{x}; \theta) \triangleq \mathbf{J}_g^{-1}(\boldsymbol{x}; \theta_1) \partial_i \mathbf{J}_g(\boldsymbol{x}; \theta)$. We first demonstrate that $\mathbf{M}$ is Lipschitz continuous:

$$
\|\mathbf{M}(i, \boldsymbol{x}; \theta_1) - \mathbf{M}(i, \boldsymbol{x}; \theta_2)\|_2
$$

$$
= \left\| \mathbf{J}_g^{-1}(\boldsymbol{x}; \theta_1) \partial_i \mathbf{J}_g(\boldsymbol{x}; \theta_1) - \mathbf{J}_g^{-1}(\boldsymbol{x}; \theta_2) \partial_i \mathbf{J}_g(\boldsymbol{x}; \theta_2) \right\|_2
$$

$$
\overset{(i)}{\leq} \left\| \mathbf{J}_g^{-1}(\boldsymbol{x}; \theta_1) \right\|_2 \|(\partial_i \mathbf{J}_g(\boldsymbol{x}; \theta_1) - \partial_i \mathbf{J}_g(\boldsymbol{x}; \theta_2))\|_2 + \left\| \mathbf{J}_g^{-1}(\boldsymbol{x}; \theta_1) - \mathbf{J}_g^{-1}(\boldsymbol{x}; \theta_2) \right\|_2 \|\partial_i \mathbf{J}_g(\boldsymbol{x}; \theta_2)\|_2
$$

$$
\overset{(ii)}{\leq} l_1'(\boldsymbol{x}) r_2(\boldsymbol{x}) \|\theta_1 - \theta_2\|_2 + l_2(\boldsymbol{x}) r_1'(\boldsymbol{x}) \|\theta_1 - \theta_2\|_2
$$

$$
= \left( l_1'(\boldsymbol{x}) r_2(\boldsymbol{x}) + l_2(\boldsymbol{x}) r_1'(\boldsymbol{x}) \right) \|\theta_1 - \theta_2\|_2,
$$

where $(i)$ is obtained by an analogous derivation of the step 1 in Lemma A.9, and $(ii)$ holds by the listed assumption.

The Lipschitzness of $\mathbf{M}$ leads to the Lipschitzness of $\frac{\partial}{\partial \boldsymbol{x}} \log |\det \mathbf{J}_g(\boldsymbol{x}; \theta)|$, since:

$$
\left\| \frac{\partial}{\partial \boldsymbol{x}} \log |\det \mathbf{J}_g(\boldsymbol{x}; \theta_1)| - \frac{\partial}{\partial \boldsymbol{x}} \log |\det \mathbf{J}_g(\boldsymbol{x}; \theta_2)| \right\|_2
$$

$$
= \|\boldsymbol{v}(\boldsymbol{x}; \theta_1) - \boldsymbol{v}(\boldsymbol{x}; \theta_2)\|_2
$$

$$
= \sqrt{\sum_i \left( \mathrm{Tr}\left( \mathbf{M}(i, \boldsymbol{x}; \theta_1) \right) - \mathrm{Tr}\left( \mathbf{M}(i, \boldsymbol{x}; \theta_2) \right) \right)^2}
$$

$$
= \sqrt{\sum_i \left( \mathrm{Tr}\left( \mathbf{M}(i, \boldsymbol{x}; \theta_1) - \mathbf{M}(i, \boldsymbol{x}; \theta_2) \right) \right)^2}
$$

$$
\overset{(i)}{\leq} \sqrt{\sum_i D^2 \|\mathbf{M}(i, \boldsymbol{x}; \theta_1) - \mathbf{M}(i, \boldsymbol{x}; \theta_2)\|_2^2}
$$

$$
\overset{(ii)}{\leq} \sqrt{\sum_i D^2 \left( l_1'(\boldsymbol{x}) r_2(\boldsymbol{x}) + l_2(\boldsymbol{x}) r_1'(\boldsymbol{x}) \right)^2 \|\theta_1 - \theta_2\|_2^2}
$$

$$
= \sqrt{D^3} \left( l_1'(\boldsymbol{x}) r_2(\boldsymbol{x}) + l_2(\boldsymbol{x}) r_1'(\boldsymbol{x}) \right) \|\theta_1 - \theta_2\|_2,
$$

where $(i)$ holds by Von Neumann's trace inequality, $(ii)$ is due to the Lipschitzness of $\mathbf{M}$.

**Step 3.** (Sufficient condition of the Lipschitzness of $\partial_i s(\boldsymbol{x};\theta)$)

$\partial_i s(\boldsymbol{x};\theta)$ can be decomposed as $(\partial_i s_{\mathbf{u}}(g(\boldsymbol{x};\theta)))^T \mathbf{J}_g(\boldsymbol{x};\theta)$, $(s_{\mathbf{u}}(g(\boldsymbol{x};\theta)))^T \partial_i \mathbf{J}_g(\boldsymbol{x};\theta)$, and $\partial_i [\boldsymbol{v}(\boldsymbol{x};\theta)]$ as follows:

$$\partial_i s(\boldsymbol{x};\theta) = \partial_i \left[ (s_{\mathbf{u}}(g(\boldsymbol{x};\theta)))^T \mathbf{J}_g(\boldsymbol{x};\theta) \right] + \partial_i [\boldsymbol{v}(\boldsymbol{x};\theta)]$$
$$= \left[ (\partial_i s_{\mathbf{u}}(g(\boldsymbol{x};\theta)))^T \mathbf{J}_g(\boldsymbol{x};\theta) \right] + \left[ (s_{\mathbf{u}}(g(\boldsymbol{x};\theta)))^T \partial_i \mathbf{J}_g(\boldsymbol{x};\theta) \right] + \partial_i [\boldsymbol{v}(\boldsymbol{x};\theta)].$$

(3.1) The Lipschitzness of $(\partial_i s_{\mathbf{u}}(g(\boldsymbol{x};\theta)))^T \mathbf{J}_g(\boldsymbol{x};\theta)$ and $(s_{\mathbf{u}}(g(\boldsymbol{x};\theta)))^T \partial_i \mathbf{J}_g(\boldsymbol{x};\theta)$ can be derived using proofs similar to that in Step 2.1:

$$\left\| (\partial_i s_{\mathbf{u}}(g(\boldsymbol{x};\theta_1)))^T \mathbf{J}_g(\boldsymbol{x};\theta_1) - (\partial_i s_{\mathbf{u}}(g(\boldsymbol{x};\theta_2)))^T \mathbf{J}_g(\boldsymbol{x};\theta_2) \right\|_2 \leq (t_2 r_1(\boldsymbol{x}) + t_4 r_0(\boldsymbol{x}) l_1(\boldsymbol{x})) \|\theta_1 - \theta_2\|_2 ,$$

$$\left\| (s_{\mathbf{u}}(g(\boldsymbol{x};\theta_1)))^T \partial_i \mathbf{J}_g(\boldsymbol{x};\theta_1) - (s_{\mathbf{u}}(g(\boldsymbol{x};\theta_2)))^T \partial_i \mathbf{J}_g(\boldsymbol{x};\theta_2) \right\|_2 \leq (t_1 r_2(\boldsymbol{x}) + t_3 r_0(\boldsymbol{x}) l_2(\boldsymbol{x})) \|\theta_1 - \theta_2\|_2 .$$

(3.2) Lipschitzness of $\partial_i [\boldsymbol{v}(\boldsymbol{x};\theta)]$:

Let $\partial_i [\boldsymbol{v}_j(\boldsymbol{x};\theta)] \triangleq \partial_i \text{Tr}(\mathbf{M}(j,\boldsymbol{x};\theta)) = \text{Tr}(\partial_i \mathbf{M}(j,\boldsymbol{x};\theta))$. We first show that $\partial_i \mathbf{M}(j,\boldsymbol{x};\theta)$ can be decomposed as:

$$\partial_i \mathbf{M}(j,\boldsymbol{x};\theta) = \partial_i \left( \mathbf{J}_g^{-1}(\boldsymbol{x};\theta) \partial_j \mathbf{J}_g(\boldsymbol{x};\theta) \right) = \left( \partial_i \mathbf{J}_g^{-1}(\boldsymbol{x};\theta) \partial_j \mathbf{J}_g(\boldsymbol{x};\theta) \right) + \left( \mathbf{J}_g^{-1}(\boldsymbol{x};\theta) \partial_i \partial_j \mathbf{J}_g(\boldsymbol{x};\theta) \right)$$

The Lipschitz constant of $\partial_i \mathbf{M}$ equals to $\left( l_2'(\boldsymbol{x}) r_2(\boldsymbol{x}) + l_2(\boldsymbol{x}) r_2'(\boldsymbol{x}) \right) + \left( l_1'(\boldsymbol{x}) r_3(\boldsymbol{x}) + l_3(\boldsymbol{x}) r_1'(\boldsymbol{x}) \right)$ based on a similar derivation as in Step 3.1. The Lipschitzness of $\partial_i \mathbf{M}(j,\boldsymbol{x};\theta)$ leads to the Lipschitzness of $\partial_i [\boldsymbol{v}(\boldsymbol{x};\theta)]$:

$$\|\partial_i [\boldsymbol{v}(\boldsymbol{x};\theta_1)] - \partial_i [\boldsymbol{v}(\boldsymbol{x};\theta_2)]\|_2$$
$$= \sqrt{\sum_j \left( \text{Tr}(\partial_i \mathbf{M}(j,\boldsymbol{x};\theta_1)) - \text{Tr}(\partial_i \mathbf{M}(j,\boldsymbol{x};\theta_2)) \right)^2}$$
$$= \sqrt{\sum_j \text{Tr}(\partial_i \mathbf{M}(j,\boldsymbol{x};\theta_1) - \partial_i \mathbf{M}(j,\boldsymbol{x};\theta_2))^2}$$
$$\overset{(i)}{\leq} \sqrt{\sum_j D^2 \|\partial_i \mathbf{M}(j,\boldsymbol{x};\theta_1) - \partial_i \mathbf{M}(j,\boldsymbol{x};\theta_2)\|_2^2}$$
$$\overset{(ii)}{\leq} \sqrt{\sum_j D^2 \left( l_2'(\boldsymbol{x}) r_2(\boldsymbol{x}) + l_2(\boldsymbol{x}) r_2'(\boldsymbol{x}) + l_1'(\boldsymbol{x}) r_3(\boldsymbol{x}) + l_3(\boldsymbol{x}) r_1'(\boldsymbol{x}) \right)^2 \|\theta_1 - \theta_2\|_2^2}$$
$$= \sqrt{D^3} \left( l_2'(\boldsymbol{x}) r_2(\boldsymbol{x}) + l_2(\boldsymbol{x}) r_2'(\boldsymbol{x}) + l_1'(\boldsymbol{x}) r_3(\boldsymbol{x}) + l_3(\boldsymbol{x}) r_1'(\boldsymbol{x}) \right) \|\theta_1 - \theta_2\|_2$$

where $(i)$ holds by Von Neumann's trace inequality, $(ii)$ is due to the Lipschitzness of $\partial_i \mathbf{M}$. $\qquad \square$

### A.1.2 Derivation of Eqs. (8) and (9)

Energy-based models are formulated based on the observation that any continuous pdf $p(\boldsymbol{x};\theta)$ can be expressed as a Boltzmann distribution $\exp(-E(\boldsymbol{x};\theta)) Z^{-1}(\theta)$ [13], where the energy function $E(\cdot;\theta)$ can be modeled as any scalar-valued continuous function. In EBFlow, the energy function $E(\boldsymbol{x};\theta)$ is selected as $-\log(p_{\mathbf{u}}(g(\boldsymbol{x};\theta)) \prod_{g_i \in \mathcal{S}_n} |\det(\mathbf{J}_{g_i}(\boldsymbol{x}_{i-1};\theta))|)$ according to Eq. (9). This suggests that the normalizing constant $Z(\theta) = \int \exp(-E(\boldsymbol{x};\theta)) d\boldsymbol{x}$ is equal to $(\prod_{g_i \in \mathcal{S}_l} |\det(\mathbf{J}_{g_i}(\theta))|)^{-1}$ according to Lemma A.11.

**Lemma A.11.**

$$\left( \prod_{g_i \in \mathcal{S}_l} |\det(\mathbf{J}_{g_i}(\theta))| \right)^{-1} = \int_{\boldsymbol{x} \in \mathbb{R}^D} p_{\mathbf{u}}(g(\boldsymbol{x};\theta)) \prod_{g_i \in \mathcal{S}_n} |\det(\mathbf{J}_{g_i}(\boldsymbol{x}_{i-1};\theta))| \, d\boldsymbol{x}. \qquad (A1)$$

*Proof.*

$$1 = \int_{\boldsymbol{x}\in\mathbb{R}^D} p(\boldsymbol{x};\theta)d\boldsymbol{x}$$

$$= \int_{\boldsymbol{x}\in\mathbb{R}^D} p_{\mathbf{u}}\left(g(\boldsymbol{x};\theta)\right) \prod_{g_i\in\mathcal{S}_n} \left|\det(\mathbf{J}_{g_j}(\boldsymbol{x}_{i-1};\theta))\right| \prod_{g_i\in\mathcal{S}_l} \left|\det(\mathbf{J}_{g_i}(\theta))\right| d\boldsymbol{x}$$

$$= \prod_{g_i\in\mathcal{S}_l} \left|\det(\mathbf{J}_{g_i}(\theta))\right| \int_{\boldsymbol{x}\in\mathbb{R}^D} p_{\mathbf{u}}\left(g(\boldsymbol{x};\theta)\right) \prod_{g_i\in\mathcal{S}_n} \left|\det(\mathbf{J}_{g_i}(\boldsymbol{x}_{i-1};\theta))\right| d\boldsymbol{x}$$

By multiplying $\left(\prod_{g_i\in\mathcal{S}_l} \left|\det(\mathbf{J}_{g_i}(\theta))\right|\right)^{-1}$ to both sides of the equation, we arrive at the conclusion:

$$\left(\prod_{g_i\in\mathcal{S}_l} \left|\det(\mathbf{J}_{g_i}(\theta))\right|\right)^{-1} = \int_{\boldsymbol{x}\in\mathbb{R}^D} p_{\mathbf{u}}\left(g(\boldsymbol{x};\theta)\right) \prod_{g_i\in\mathcal{S}_n} \left|\det(\mathbf{J}_{g_i}(\boldsymbol{x}_{i-1};\theta))\right| d\boldsymbol{x}.$$

$\square$

Figure A1: An illustration of the relationship between the variables discussed in Proposition 4.1, Lemma A.12, and Lemma A.13. $\mathbf{x}$ represents a random vector sampled from the data distribution $p_{\mathbf{x}}$. $\{g_i\}_{i=1}^L$ is a series of transformations. $\mathbf{x}_j \triangleq g_j \circ \cdots \circ g_1(\mathbf{x})$, and $p_{\mathbf{x}_j}$ is its pdf. $p_j(\boldsymbol{x}_j) = p_{\mathbf{u}}(g_L \circ \cdots \circ g_{j+1}(\boldsymbol{x}_j)) \prod_{i=j+1}^L |\det(\mathbf{J}_{g_i})|$, where $p_{\mathbf{u}}$ is a prior distribution. The properties of KL divergence and Fisher divergence presented in the last two rows are derived in Lemmas A.12 and A.13.

### A.1.3 Theoretical Analyses of KL Divergence and Fisher Divergence

In this section, we provide formal derivations for Proposition 4.1, Lemma A.12, and Lemma A.13. To ensure a clear presentation, we provide a visualization of the relationship between the variables used in the subsequent derivations in Fig. A1.

**Lemma A.12.** *Let $p_{\mathbf{x}_j}$ be the pdf of the latent variable of $\mathbf{x}_j \triangleq g_j\circ\cdots\circ g_1(\mathbf{x})$ indexed by $j$. In addition, let $p_j(\cdot)$ be a pdf modeled as $p_{\mathbf{u}}(g_L \circ \cdots \circ g_{j+1}(\cdot)) \prod_{i=j+1}^L |\det(\mathbf{J}_{g_i})|$, where $j \in \{0,\cdots,L-1\}$. It follows that:*

$$\mathbb{D}_{\mathrm{KL}}\left[p_{\mathbf{x}_j}\|p_j\right] = \mathbb{D}_{\mathrm{KL}}\left[p_{\mathbf{x}}\|p_0\right], \forall j \in \{1,\cdots,L-1\}. \tag{A2}$$

*Proof.* The equivalence $\mathbb{D}_{\mathrm{KL}}\left[p_{\mathbf{x}}\|p_0\right] = \mathbb{D}_{\mathrm{KL}}\left[p_{\mathbf{x}_j}\|p_j\right]$ holds for any $j \in \{1, \cdots, L-1\}$ since:

$$
\begin{aligned}
&\mathbb{D}_{\mathrm{KL}}\left[p_{\mathbf{x}}\|p_0\right] \\
&= \mathbb{E}_{p_{\mathbf{x}}(\boldsymbol{x})}\left[\log\left(\frac{p_{\mathbf{x}}(\boldsymbol{x})}{p_0(\boldsymbol{x})}\right)\right] \\
&= \mathbb{E}_{p_{\mathbf{x}}(\boldsymbol{x})}\left[\log\left(\frac{p_{\mathbf{x}_j}(g_j \circ \cdots \circ g_1(\boldsymbol{x}))\prod_{i=1}^{j}|\det(\mathbf{J}_{g_i})|}{p_{\mathbf{u}}(g_L \circ \cdots \circ g_1(\boldsymbol{x}))\prod_{i=1}^{L}|\det(\mathbf{J}_{g_i})|}\right)\right] \\
&= \mathbb{E}_{p_{\mathbf{x}}(\boldsymbol{x})}\left[\log\left(\frac{p_{\mathbf{x}_j}(g_j \circ \cdots \circ g_1(\boldsymbol{x}))}{p_{\mathbf{u}}(g_L \circ \cdots \circ g_1(\boldsymbol{x}))\prod_{i=j+1}^{L}|\det(\mathbf{J}_{g_i})|}\right)\right] \\
&\overset{(i)}{=} \mathbb{E}_{p_{\mathbf{x}_j}(\boldsymbol{x}_j)}\left[\log\left(\frac{p_{\mathbf{x}_j}(\boldsymbol{x}_j)}{p_{\mathbf{u}}(g_L \circ \cdots \circ g_{j+1}(\boldsymbol{x}_j))\prod_{i=j+1}^{L}|\det(\mathbf{J}_{g_i})|}\right)\right] \\
&= \mathbb{D}_{\mathrm{KL}}\left[p_{\mathbf{x}_j}\|p_j\right],
\end{aligned}
$$

where $(i)$ is due to the property that $\mathbb{E}_{p_{\mathbf{x}}(\boldsymbol{x})}[f \circ g_j \circ \cdots \circ g_1(\boldsymbol{x})] = \mathbb{E}_{p_{\mathbf{x}_j}(\boldsymbol{x}_j)}[f(\boldsymbol{x}_j)]$ for a given function $f$. Therefore, $\mathbb{D}_{\mathrm{KL}}\left[p_{\mathbf{x}_j}\|p_j\right] = \mathbb{D}_{\mathrm{KL}}\left[p_{\mathbf{x}}\|p_0\right], \forall j \in \{1, \cdots, L-1\}$. $\qquad\square$

**Lemma A.13.** *Let $p_{\mathbf{x}_j}$ be the pdf of the latent variable of $\mathbf{x}_j \triangleq g_j \circ \cdots \circ g_1(\mathbf{x})$ indexed by $j$. In addition, let $p_j(\cdot)$ be a pdf modeled as $p_{\mathbf{u}}(g_L \circ \cdots \circ g_{j+1}(\cdot))\prod_{i=j+1}^{L}|\det(\mathbf{J}_{g_i})|$, where $j \in \{0, \cdots, L-1\}$. It follows that:*

$$
\mathbb{D}_{\mathrm{F}}\left[p_{\mathbf{x}}\|p_0\right] = \mathbb{E}_{p_{\mathbf{x}_j}(\boldsymbol{x}_j)}\left[\frac{1}{2}\left\|\left(\frac{\partial}{\partial \boldsymbol{x}_j}\log\left(\frac{p_{\mathbf{x}_j}(\boldsymbol{x}_j)}{p_j(\boldsymbol{x}_j)}\right)\right)\prod_{i=1}^{j}\mathbf{J}_{g_i}\right\|^2\right], \forall j \in \{1, \cdots, L-1\}. \quad \text{(A3)}
$$

*Proof.* Based on the definition, the Fisher divergence between $p_{\mathbf{x}}$ and $p_0$ is written as:

$$
\begin{aligned}
&\mathbb{D}_{\mathrm{F}}\left[p_{\mathbf{x}}\|p_0\right] \\
&= \mathbb{E}_{p_{\mathbf{x}}(\boldsymbol{x})}\left[\frac{1}{2}\left\|\frac{\partial}{\partial \boldsymbol{x}}\log\left(\frac{p_{\mathbf{x}}(\boldsymbol{x})}{p_0(\boldsymbol{x})}\right)\right\|^2\right] \\
&= \mathbb{E}_{p_{\mathbf{x}}(\boldsymbol{x})}\left[\frac{1}{2}\left\|\frac{\partial}{\partial \boldsymbol{x}}\log\left(\frac{p_{\mathbf{x}_j}(g_j \circ \cdots \circ g_1(\boldsymbol{x}))\prod_{i=1}^{j}|\det(\mathbf{J}_{g_i})|}{p_{\mathbf{u}}(g_L \circ \cdots \circ g_1(\boldsymbol{x}))\prod_{i=1}^{L}|\det(\mathbf{J}_{g_i})|}\right)\right\|^2\right] \\
&= \mathbb{E}_{p_{\mathbf{x}}(\boldsymbol{x})}\left[\frac{1}{2}\left\|\frac{\partial}{\partial \boldsymbol{x}}\log\left(\frac{p_{\mathbf{x}_j}(g_j \circ \cdots \circ g_1(\boldsymbol{x}))}{p_{\mathbf{u}}(g_L \circ \cdots \circ g_1(\boldsymbol{x}))\prod_{i=j+1}^{L}|\det(\mathbf{J}_{g_i})|}\right)\right\|^2\right] \\
&= \mathbb{E}_{p_{\mathbf{x}}(\boldsymbol{x})}\left[\frac{1}{2}\left\|\left(\frac{\partial}{\partial g_j \circ \cdots \circ g_1(\boldsymbol{x})}\log\left(\frac{p_{\mathbf{x}_j}(g_j \circ \cdots \circ g_1(\boldsymbol{x}))}{p_{\mathbf{u}}(g_L \circ \cdots \circ g_1(\boldsymbol{x}))\prod_{i=j+1}^{L}|\det(\mathbf{J}_{g_i})|}\right)\right)\frac{\partial g_j \circ \cdots \circ g_1(\boldsymbol{x})}{\partial \boldsymbol{x}}\right\|^2\right] \\
&\overset{(i)}{=} \mathbb{E}_{p_{\mathbf{x}}(\boldsymbol{x})}\left[\frac{1}{2}\left\|\left(\frac{\partial}{\partial g_j \circ \cdots \circ g_1(\boldsymbol{x})}\log\left(\frac{p_{\mathbf{x}_j}(g_j \circ \cdots \circ g_1(\boldsymbol{x}))}{p_{\mathbf{u}}(g_L \circ \cdots \circ g_1(\boldsymbol{x}))\prod_{i=j+1}^{L}|\det(\mathbf{J}_{g_i})|}\right)\right)\prod_{i=1}^{j}\mathbf{J}_{g_i}\right\|^2\right] \\
&\overset{(ii)}{=} \mathbb{E}_{p_{\mathbf{x}_j}(\boldsymbol{x}_j)}\left[\frac{1}{2}\left\|\left(\frac{\partial}{\partial \boldsymbol{x}_j}\log\left(\frac{p_{\mathbf{x}_j}(\boldsymbol{x}_j)}{p_{\mathbf{u}}(g_L \circ \cdots \circ g_{j+1}(\boldsymbol{x}_j))\prod_{i=j+1}^{L}|\det(\mathbf{J}_{g_i})|}\right)\right)\prod_{i=1}^{j}\mathbf{J}_{g_i}\right\|^2\right], \\
&= \mathbb{E}_{p_{\mathbf{x}_j}(\boldsymbol{x}_j)}\left[\frac{1}{2}\left\|\left(\frac{\partial}{\partial \boldsymbol{x}_j}\log\left(\frac{p_{\mathbf{x}_j}(\boldsymbol{x}_j)}{p_j(\boldsymbol{x}_j)}\right)\right)\prod_{i=1}^{j}\mathbf{J}_{g_i}\right\|^2\right],
\end{aligned}
$$

where $(i)$ is due to the chain rule, and $(ii)$ is because $\mathbb{E}_{p_{\mathbf{x}}(\boldsymbol{x})}[f \circ g_j \circ \cdots \circ g_1(\boldsymbol{x})] = \mathbb{E}_{p_{\mathbf{x}_j}(\boldsymbol{x}_j)}[f(\boldsymbol{x}_j)]$ for a given function $f$. $\qquad\square$

*Remark* A.14. Lemma A.13 implies that $\mathbb{D}_F\left[p_{\mathbf{x}_j}\|p_j\right] \neq \mathbb{D}_F\left[p_{\mathbf{x}}\|p_0\right]$ in general, as the latter contains an additional multiplier $\prod_{i=1}^j \mathbf{J}_{g_i}$ as shown below:

$$\mathbb{D}_F\left[p_{\mathbf{x}}\|p_0\right] = \mathbb{E}_{p_{\mathbf{x}_j}(\boldsymbol{x}_j)}\left[\frac{1}{2}\left\|\left(\frac{\partial}{\partial \boldsymbol{x}_j}\log\left(\frac{p_{\mathbf{x}_j}(\boldsymbol{x}_j)}{p_j(\boldsymbol{x}_j)}\right)\right)\prod_{i=1}^j \mathbf{J}_{g_i}\right\|^2\right],$$

$$\mathbb{D}_F\left[p_{\mathbf{x}_j}\|p_j\right] = \mathbb{E}_{p_{\mathbf{x}_j}(\boldsymbol{x}_j)}\left[\frac{1}{2}\left\|\left(\frac{\partial}{\partial \boldsymbol{x}_j}\log\left(\frac{p_{\mathbf{x}_j}(\boldsymbol{x}_j)}{p_j(\boldsymbol{x}_j)}\right)\right)\right\|^2\right].$$

**Proposition 4.1.** *Let $p_{\mathbf{x}_j}$ be the pdf of the latent variable of $\mathbf{x}_j \triangleq g_j \circ \cdots \circ g_1(\mathbf{x})$ indexed by $j$. In addition, let $p_j(\cdot)$ be a pdf modeled as $p_{\mathbf{u}}(g_L \circ \cdots \circ g_{j+1}(\cdot))\prod_{i=j+1}^L |\det(\mathbf{J}_{g_i})|$, where $j \in \{0, \cdots, L-1\}$. It follows that:*

$$\mathbb{D}_F\left[p_{\mathbf{x}_j}\|p_j\right] = 0 \Leftrightarrow \mathbb{D}_F\left[p_{\mathbf{x}}\|p_0\right] = 0, \forall j \in \{1, \cdots, L-1\}. \tag{A4}$$

*Proof.* Based on Remark A.14, the following holds:

$$\mathbb{D}_F\left[p_{\mathbf{x}_j}\|p_j\right] = \mathbb{E}_{p_{\mathbf{x}_j}(\boldsymbol{x}_j)}\left[\frac{1}{2}\left\|\frac{\partial}{\partial \boldsymbol{x}_j}\log\left(\frac{p_{\mathbf{x}_j}(\boldsymbol{x}_j)}{p_j(\boldsymbol{x}_j)}\right)\right\|^2\right] = 0$$

$$\overset{(i)}{\Leftrightarrow} \left\|\frac{\partial}{\partial \boldsymbol{x}_j}\log\left(\frac{p_{\mathbf{x}_j}(\boldsymbol{x}_j)}{p_j(\boldsymbol{x}_j)}\right)\right\|^2 = 0$$

$$\overset{(ii)}{\Leftrightarrow} \left\|\frac{\partial}{\partial \boldsymbol{x}_j}\log\left(\frac{p_{\mathbf{x}_j}(\boldsymbol{x}_j)}{p_j(\boldsymbol{x}_j)}\right)\prod_{i=1}^j \mathbf{J}_{g_i}\right\|^2 = 0$$

$$\overset{(i)}{\Leftrightarrow} \mathbb{D}_F\left[p_{\mathbf{x}}\|p_0\right] = \mathbb{E}_{p_{\mathbf{x}_j}(\boldsymbol{x}_j)}\left[\frac{1}{2}\left\|\left(\frac{\partial}{\partial \boldsymbol{x}_j}\log\left(\frac{p_{\mathbf{x}_j}(\boldsymbol{x}_j)}{p_j(\boldsymbol{x}_j)}\right)\right)\prod_{i=1}^j \mathbf{J}_{g_i}\right\|^2\right] = 0,$$

where $(i)$ and $(ii)$ both result from the positiveness condition presented in Assumption A.1. Specifically, for $(i)$, $p_{\mathbf{x}_j}(\boldsymbol{x}_j) = p_{\mathbf{x}}(g_1^{-1} \circ \cdots \circ g_j^{-1}(\boldsymbol{x}_j))\prod_{i=1}^j \left|\det\left(\mathbf{J}_{g_i^{-1}}\right)\right| > 0$, since $p_{\mathbf{x}} > 0$ and $\prod_{i=1}^j \left|\det\left(\mathbf{J}_{g_i^{-1}}\right)\right| = \prod_{i=1}^j |\det(\mathbf{J}_{g_i})|^{-1} > 0$. Meanwhile $(ii)$ holds since $\prod_{i=1}^j |\det(\mathbf{J}_{g_i})| > 0$ and thus all of the singular values of $\prod_{i=1}^j \mathbf{J}_{g_i}$ are non-zero. $\square$

## A.2 Experimental Setups

In this section, we elaborate on the experimental setups and provide the detailed configurations for the experiments presented in Section 5 of the main manuscript. The code implementation for the experiments is provided in the following repository: `https://github.com/chen-hao-chao/ebflow`. Our code implementation is developed based on [7, 17, 44].

### A.2.1 Experimental Setups for the Two-Dimensional Synthetic Datasets

**Datasets.** In Section 5.1, we present the experimental results on three two-dimensional synthetic datasets: Sine, Swirl, and Checkerboard. The Sine dataset is generated by sampling data points from the set $\{(4w - 2, \sin(12w - 6)) \mid w \in [0, 1]\}$. The Swirl dataset is generated by sampling data points from the set $\{(-\pi\sqrt{w}\cos(\pi\sqrt{w}), \pi\sqrt{w}\sin(\pi\sqrt{w})) \mid w \in [0, 1]\}$. The Checkerboard dataset is generated by sampling data points from the set $\{(4w - 2, t - 2s + \lfloor 4w - 2\rfloor \bmod 2) \mid w \in [0, 1], t \in [0, 1], s \in \{0, 1\}\}$, where $\lfloor \cdot \rfloor$ is a floor function, and $\bmod$ represents the modulo operation.

To establish $p_{\mathbf{x}}$ for all three datasets, we smooth a Dirac function using a Gaussian kernel. Specifically, we define the Dirac function as $\hat{p}(\hat{\boldsymbol{x}}) \triangleq \frac{1}{M}\sum_{i=1}^M \delta(\|\hat{\boldsymbol{x}} - \hat{\boldsymbol{x}}^{(i)}\|)$, where $\{\hat{\boldsymbol{x}}^{(i)}\}_{i=1}^M$ are $M$ uniformly-sampled data points. The data distribution is defined as $p_{\mathbf{x}}(\boldsymbol{x}) \triangleq \int \hat{p}(\hat{\boldsymbol{x}})\mathcal{N}(\boldsymbol{x}|\hat{\boldsymbol{x}}, \hat{\sigma}^2\boldsymbol{I})d\hat{\boldsymbol{x}} = \frac{1}{M}\sum_{i=1}^M \mathcal{N}(\boldsymbol{x}|\hat{\boldsymbol{x}}^{(i)}, \hat{\sigma}^2\boldsymbol{I})$. The closed-form expressions for $p_{\mathbf{x}}(\boldsymbol{x})$ and $\frac{\partial}{\partial \boldsymbol{x}}\log p_{\mathbf{x}}(\boldsymbol{x})$ can be obtained using the derivation in [45]. In the experiments, $M$ is set as $50,000$, and $\hat{\sigma}$ is fixed at $0.375$ for all three datasets.

**Implementation Details.** The model architecture of $g(\cdot\,;\theta)$ consists of ten Glow blocks [4]. Each block comprises an actnorm [4] layer, a fully-connected layer, and an affine coupling layer. Table A2 provides the formal definitions of these operations. $p_{\mathbf{u}}(\cdot)$ is implemented as an isotropic Gaussian with zero mean and unit variance. To determine the best hyperparameters, we perform a grid search over the following optimizers, learning rates, and gradient clipping values based on the evaluation results in terms of the KL divergence. The optimizers include Adam [46], AdamW [47], and RMSProp. The learning rate and gradient clipping values are selected from (5e-3, 1e-3, 5e-4, 1e-4) and (None, 2.5, 10.0), respectively. Table A1 summarizes the selected hyperparameters. The optimization processes of Sine and Swirl datasets require 50,000 training iterations for convergence, while that of the Checkerboard dataset requires 100,000 iterations. The batch size is fixed at 5,000 for all setups.

### A.2.2 Experimental Setups for the Real-world Datasets

**Datasets.** The experiments presented in Section 5.2 are performed on the MNIST [19] and CIFAR-10 [37] datasets. The training and test sets of MNIST and CIFAR-10 contain 50,000 and 10,000 images, respectively. The data are smoothed using the uniform dequantization method presented in [1]. The observable parts (i.e., $\boldsymbol{x}_O$) of the images in Fig. 5 are produced using the pre-trained model in [48].

**Implementation Details.** In Sections 5.2 and 5.4, we adopt three types of model architectures: FC-based [7], CNN-based, and Glow [4] models. The FC-based model contains two fully-connected layers and a smoothed leaky ReLU non-linearity [7] in between, which is identical to [7]. The CNN-based model consists of three convolutional blocks and two squeezing operations [2] between every convolutional block. Each convolutional block contains two convolutional layers and a smoothed leaky ReLU in between. The Glow model adopted in Section 5.4 is composed of 16 Glow blocks. Each of the Glow block consists of an actnorm [4] layer, a convolutional layer, and an affine coupling layer. The squeezing operation is inserted between every eight blocks. The operations used in these models are summarized in Table A2. The smoothness factor $\alpha$ of Smooth Leaky ReLU is set to 0.3 and 0.6 for models trained on MNIST and CIFAR-10, respectively. The scaling and transition functions $s(\cdot\,;\theta)$ and $t(\cdot\,;\theta)$ of the affine coupling layers are convolutional blocks with ReLU activation functions. The prior distribution $p_{\mathbf{u}}(\cdot)$ is implemented as an isotropic Gaussian with zero mean and unit variance. The FC-based and CNN-based models are trained with RMSProp using a learning rate initialized at 1e-4 and a batch size of 100. The Glow model is trained with an Adam optimizer using a learning rate initialized at 1e-4 and a batch size of 100. The gradient clipping value is set to 500 during the training for the Glow model. The learning rate scheduler `MultiStepLR` in `PyTorch` is used for gradually decreasing the learning rates. The hyper-parameters $\{\sigma, \xi\}$ used in DSM and FDSSM are selected based on a grid search over $\{0.05, 0.1, 0.5, 1.0\}$. The selected $\{\sigma, \xi\}$ are $\{1.0, 1.0\}$ and $\{0.1, 0.1\}$ for the MNIST and CIFAR-10 datasets, respectively. The parameter $m$ in EMA is set to 0.999. The algorithms are implemented using `PyTorch` [39]. The gradients w.r.t. $\boldsymbol{x}$ and $\theta$ are both calculated using automatic differential tools [40] provided by `PyTorch` [39]. The runtime is evaluated on Tesla V100 NVIDIA GPUs. In the experiments performed on CIFAR-10 and CelebA using score-matching methods, the energy function (i.e., $\mathbb{E}_{p_{\mathbf{x}}(\boldsymbol{x})}\left[E(\boldsymbol{x};\theta)\right]$) is added as a regularization loss with a balancing factor fixed at 0.001 during the optimization processes. The results in Fig. 2 (b) are smoothed with the exponential moving average function used in `Tensorboard` [49], i.e., $w \times d_{i-1} + (1-w) \times d_i$, where $w$ is set to 0.45 and $d_i$ represents the evaluation result at the $i$-th iteration.

Table A1: The hyper-parameters used in the two-dimensional synthetic example in Section 5.1.

| Dataset | | ML | SML | SSM | DSM | FDSSM |
|---|---|---|---|---|---|---|
| | Optimizor | Adam | AdamW | Adam | Adam | Adam |
| Sine | Learning Rate | 5e-4 | 5e-4 | 1e-4 | 1e-4 | 1e-4 |
| | Gradient Clip | 1.0 | None | 1.0 | 1.0 | 1.0 |
| | Optimizor | Adam | Adam | Adam | Adam | Adam |
| Swirl | Learning Rate | 5e-3 | 1e-4 | 1e-4 | 1e-4 | 1e-4 |
| | Gradient Clip | None | 10.0 | 10.0 | 10.0 | 2.5 |
| | Optimizor | AdamW | AdamW | AdamW | AdamW | Adam |
| Checkerboard | Learning Rate | 1e-4 | 1e-4 | 1e-4 | 1e-4 | 1e-4 |
| | Gradient Clip | 10.0 | 10.0 | 10.0 | 10.0 | 10.0 |

Table A2: The components of $g(\cdot;\theta)$ used in this paper. In this table, $\boldsymbol{z}$ and $\boldsymbol{y}$ are the output and the input of a layer, respectively. $\beta$ and $\gamma$ represent the mean and variance of an actnorm layer. $\boldsymbol{w}$ is a convolutional kernel, and $\boldsymbol{w} \star \boldsymbol{y} \triangleq \hat{\boldsymbol{W}}\boldsymbol{y}$, where $\star$ is a convolutional operator, and $\hat{\boldsymbol{W}}$ is a $D \times D$ matrix. $\boldsymbol{W}$ and $\boldsymbol{b}$ represent the weight and bias in a fully-connected layer. $\alpha$ is a hyper-parameter for adjusting the smoothness of smooth leaky ReLU. In the affine coupling layer, $\boldsymbol{z}$ and $\boldsymbol{y}$ are split into two parts $\{\boldsymbol{z}_a, \boldsymbol{z}_b\}$ and $\{\boldsymbol{y}_a, \boldsymbol{y}_b\}$, respectively. $s(\cdot;\theta)$ and $t(\cdot;\theta)$ are the scaling and transition networks parameterized with $\theta$. $\mathrm{sig}\,(y) = 1/(1 + \exp\,(-y))$ represents the sigmoid function. $\dim\,(\cdot)$ represents the dimension of the input vector. $\boldsymbol{y}_{[i]}$ represents the $i$-th element of vector $\boldsymbol{y}$.

| Layer | Function | Log Jacobian Determinant | Set |
|---|---|---|---|
| actnorm [4] | $\boldsymbol{z} = (\boldsymbol{y} - \beta)/\gamma$ | $\sum_{i=1}^{D} \log \left|1/\gamma_{[i]}\right|$ | $\mathcal{S}_l$ |
| convolutional | $\boldsymbol{z} = \boldsymbol{w} \star \boldsymbol{y} + \boldsymbol{b}$ | $\log\left|\det\left(\hat{\boldsymbol{W}}\right)\right|$ | $\mathcal{S}_l$ |
| fully-connected | $\boldsymbol{z} = \boldsymbol{W}\boldsymbol{y} + \boldsymbol{b}$ | $\log\left|\det\left(\boldsymbol{W}\right)\right|$ | $\mathcal{S}_l$ |
| smooth leaky ReLU [7] | $\boldsymbol{z} = \alpha\boldsymbol{y} + (1-\alpha)\log(1 + \exp\,(\boldsymbol{y}))$ | $\sum_{i=1}^{D} \log\left|\alpha + (1-\alpha)\mathrm{sig}\,(\boldsymbol{y}_{[i]})\right|$ | $\mathcal{S}_n$ |
| affine coupling [4] | $\boldsymbol{z}_a = s(\boldsymbol{y}_b;\theta)\boldsymbol{y}_a + t(\boldsymbol{y}_b;\theta),\, \boldsymbol{z}_b = \boldsymbol{y}_b$ | $\sum_{i=1}^{\dim(\boldsymbol{y}_b)} \log\left|s(\boldsymbol{y}_b;\theta)_{[i]}\right|$ | $\mathcal{S}_n$ |

Table A3: The simulation results of Eq. (A6). The error rate is measured by $|d_{\mathrm{true}} - d_{\mathrm{est}}|/|d_{\mathrm{true}}|$, where $d_{\mathrm{true}}$ and $d_{\mathrm{est}}$ represent the true and estimated Jacobian determinants, respectively.

| | $D = 50$ | $D = 100$ | $D = 200$ |
|---|---|---|---|
| Error Rate ($M = 50$) | 0.004211 | 0.099940 | 0.355314 |
| Error Rate ($M = 100$) | 0.003503 | 0.034608 | 0.076239 |
| Error Rate ($M = 200$) | **0.002332** | **0.015411** | **0.011175** |

**Results of the Related Works.** The results of the relative gradient [7], SSM [16], and FDSSM [17] methods are directly obtained from their original paper. On the other hand, the results of the DSM method is obtained from [17]. Please note that the reported results of [16] and [17] differ from each other given that they both adopt the NICE [1] model. Specifically, the SSM method achieves NLL$= 3,355$ and NLL$= 6,234$ in [16] and [17], respectively. Moreover, the DSM method achieves NLL$= 4,363$ and NLL$= 3,398$ in [16] and [17], respectively. In Table 4, we report the results with lower NLL.

### A.3 Estimating the Jacobian Determinants using Importance Sampling

Importance sampling is a technique used to estimate integrals, which can be employed to approximate the normalizing constant $Z(\theta)$ in an energy-based model. In this method, a pdf $q$ with a simple closed form that can be easily sampled from is selected. The normalizing constant can then be expressed as the following formula:

$$Z(\theta) = \int_{\boldsymbol{x}\in\mathbb{R}^D} \exp\left(-E(\boldsymbol{x};\theta)\right) d\boldsymbol{x} = \int_{\boldsymbol{x}\in\mathbb{R}^D} q(\boldsymbol{x}) \frac{\exp\left(-E(\boldsymbol{x};\theta)\right)}{q(\boldsymbol{x})} d\boldsymbol{x}$$

$$= \mathbb{E}_{q(\boldsymbol{x})}\left[\frac{\exp\left(-E(\boldsymbol{x};\theta)\right)}{q(\boldsymbol{x})}\right] \approx \frac{1}{M} \sum_{j=1}^{M} \frac{\exp\left(-E(\hat{\boldsymbol{x}}^{(j)};\theta)\right)}{q(\hat{\boldsymbol{x}}^{(j)})},$$

(A5)

where $\{\hat{\boldsymbol{x}}^{(j)}\}_{j=1}^{M}$ represents $M$ i.i.d. samples drawn from $q$. According to Lemma A.11, the Jacobian determinants of the layers in $\mathcal{S}_l$ can be approximated using Eq. (A5) as follows:

$$\left(\prod_{g_i\in\mathcal{S}_l} |\det(\mathbf{J}_{g_i}(\theta))|\right)^{-1} \approx \frac{1}{M} \sum_{j=1}^{M} \frac{p_\mathbf{u}\left(g(\hat{\boldsymbol{x}}^{(j)};\theta)\right) \prod_{g_i\in\mathcal{S}_n} \left|\det(\mathbf{J}_{g_i}(\hat{\boldsymbol{x}}_{i-1}^{(j)};\theta))\right|}{q(\hat{\boldsymbol{x}}^{(j)})}.$$

(A6)

Table A4: An overall comparison between EBFlow, the baseline method, the Relative Gradient method [7], and the methods that utilize specially designed linear layers [8–12, 29]. The notations ✓/ ✗ in row 'Unbiased' represent whether the models are optimized according to an unbiased target. On the other hand, the notations ✓/ ✗ in row 'Unconstrained' represent whether the models can be constructed with arbitrary linear transformations. [†] The approximation errors $o(\xi)$ of FDSSM is controlled by its hyper-parameter $\xi$. [‡] The error $o(\boldsymbol{W})$ of the Relative Gradient method is determined by the values of a model's weights.

| | KL-Divergence-Based | | | | Fisher-Divergence-Based | | |
| | Baseline (ML) | EBFlow (SML) | Relative Grad. | Special Linear | EBFlow (SSM) | EBFlow (DSM) | EBFlow (FDSSM) |
|---|---|---|---|---|---|---|---|
| Complexity | $\mathcal{O}(D^3L)$ | $\mathcal{O}(D^3L)$ | $\mathcal{O}(D^2L)$ | $\mathcal{O}(D^2L)$ | $\mathcal{O}(D^2L)$ | $\mathcal{O}(D^2L)$ | $\mathcal{O}(D^2L)$ |
| Unbiased | ✓ | ✓ | ✗[‡] | ✓ | ✓ | ✗ | ✗[†] |
| Unconstrained | ✓ | ✓ | ✓ | ✗ | ✓ | ✓ | ✓ |

To validate this idea, we provide a simple simulation with $p_{\mathbf{u}} = \mathcal{N}(\mathbf{0}, \boldsymbol{I})$, $q = \mathcal{N}(\mathbf{0}, \boldsymbol{I})$, $g(\boldsymbol{x}; \boldsymbol{W}) = \boldsymbol{W}\boldsymbol{x}$, $M = \{50, 100, 200\}$, and $D = \{50, 100, 200\}$ in Table A3. The results show that larger values of $M$ lead to more accurate estimation of the Jacobian determinants. Typically, the choice of $q$ is crucial to the accuracy of importance sampling. To obtain an accurate approximation, one can adopt the technique of annealed importance sampling (AIS) [33] or Reverse AIS Estimator (RAISE) [34], which are commonly-adopted algorithms for effectively estimating $Z(\theta)$.

Eq. (A6) can be interpreted as a generalization of the stochastic estimator presented in [50], where the distributions $p_{\mathbf{u}}$ and $q$ are modeled as isotropic Gaussian distributions, and $g$ is restricted as a linear transformation. For the further analysis of this concept, particularly in the context of determinant estimation for matrices, we refer readers to Section I of [50], where a more sophisticated approximation approach and the corresponding experimental findings are provided.

## A.4   A Comparison among the Methods Discussed in this Paper

In Sections 2, 3, and 4, we discuss various methods for efficiently training flow-based models. To provide a comprehensive comparison of these methods, we summarize their complexity and characteristics in Table A4.

## A.5   The Impacts of the Constraint of Linear Transformations on the Performance of a Flow-based Model

In this section, we examine the impact of the constraints of linear transformations on the performance of a flow-based model. A key distinction between constrained and unconstrained linear layers lies

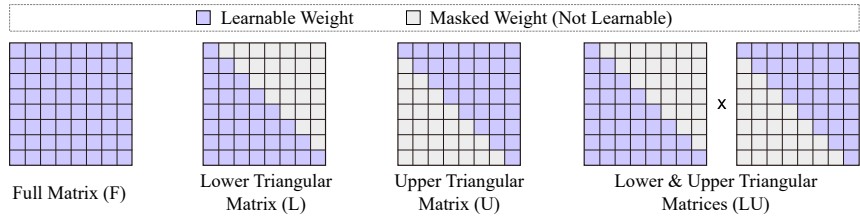

Figure A2: An illustration of the weight matrices in the F, L, U, and LU layers described in Section A.5.

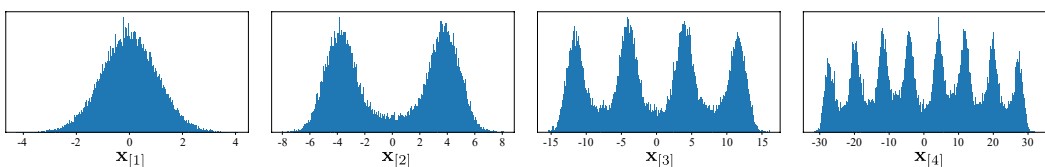

Figure A3: Visualized marginal distributions of $p_{\mathbf{x}_{[i]}}$ for $i = 1, 2, 3,$ and 4.

in how they model the correlation between each element in a data vector. Constrained linear transformations, such as those used in the previous works [8–12, 29], impose predetermined correlations that are not learnable during the optimization process. For instance, masked linear layers [8–10] are constructed by masking either the upper or lower triangular weight matrix in a linear layer. In contrast, unconstrained linear layers have weight matrices that are fully learnable, making them more flexible than their constrained counterparts.

To demonstrate the influences of the constraint on the expressiveness of a model, we provide a performance comparison between flow-based models constructed using different types of linear layers. Specifically, we compare the performance of the models constructed using linear layers with full matrices, lower triangular matrices, upper triangular matrices, and matrices that are the multiplication of both lower and upper triangular matrices. These four types of linear layers are hereafter denoted as F, L, U, and LU, respectively, and the differences between them are depicted in Fig. A2. Furthermore, to highlight the performance discrepancy between these models, we construct the target distribution $p_\mathbf{x}$ based on an autoregressive relationship of data vector $\mathbf{x}$. Let $\mathbf{x}_{[i]}$ denote the $i$-th element of $\mathbf{x}$, and $p_{\mathbf{x}_{[i]}}$ represent its associated pdf. $\mathbf{x}_{[i]}$ is constructed based on the following equation:

$$\mathbf{x}_{[i]} = \begin{cases} \mathbf{u}_{[0]} & \text{if } i = 1, \\ \tanh(\mathbf{u}_{[i]} \times s) \times (\mathbf{x}_{[i-1]} + d \times 2^i), & \text{if } i \in \{2, \ldots, D\}, \end{cases} \tag{A7}$$

where $\mathbf{u}$ is sampled from an isotropic Gaussian, and $s$ and $d$ are coefficients controlling the shape and distance between each mode, respectively. In Eq. (A7), the function $\tanh(\cdot)$ can be intuitively viewed as a smoothed variant of the function $2H(\cdot) - 1$, where $H(\cdot)$ represents the Heaviside step function. In this context, the values of $(\mathbf{x}_{[i-1]} + d \times 2^i)$ are multiplied by a value close to either $-1$ or 1, effectively transforming a positive number to a negative one. Fig. A3 depicts a number of examples of $p_{\mathbf{x}_{[i]}}$ constructed using this method. By employing this approach to design $p_\mathbf{x}$, where capturing $p_{\mathbf{x}_{[i]}}$ is presumed to be more challenging than modeling $p_{\mathbf{x}_{[j]}}$ for any $j < i$, we can inspect how the applied constraints impact performance. Inappropriately masking the linear layers, like the U-type layer, is anticipated to result in degraded performance, similar to the *anti-casual* effect explained in [51].

In this experiment, we constructed flow-based models using the smoothed leakyReLU activation and different types of linear layers (i.e., F, L, U, and LU) with a dimensionality of $D = 10$. The models are optimized according to Eq. (2). The performance of these models is evaluated in terms of NLL, and its trends are depicted in Fig. A4. It is observed that the flow-based model built with the F-type layers achieved the lowest NLL, indicating the advantage of using unconstrained weight matrices in linear layers. In addition, there is a noticeable performance discrepancy between models with the L-type and U-type layers, indicating that imposing inappropriate constraints on linear layers may negatively affect the modeling abilities of flow-based mod-

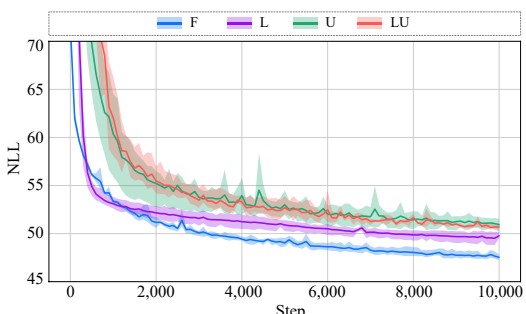

Figure A4: The evaluation curves in terms of NLL of the flow-based models constructed with the F-type, L-type, U-type, and LU-type layers. The curves and shaded area depict the mean and 95% confidence interval of three independent runs.

els. Furthermore, even when both L-type and U-type layers were adopted, as shown in the red curve in Fig. A4, the performance remains inferior to those using the F-type layers. This experimental evidence suggests that linear layers constructed based on matrix decomposition (e.g., [4, 9]) may not possess the same expressiveness as unconstrained linear layers.

## A.6    Limitations and Discussions

We noticed that score-matching methods sometimes exhibit difficulty in differentiating the weights between individual modes within a multi-modal distribution. This deficiency is illustrated in Fig. A5 (a), where EBFlow fails to accurately capture the density of the Checkerboard dataset. This phenomenon bears resemblance to the *blindness* problem discussed in [52]. While the solution proposed in [52] has the potential to address this issue, their approach is not directly applicable to the flow-based architectures employed in this paper.

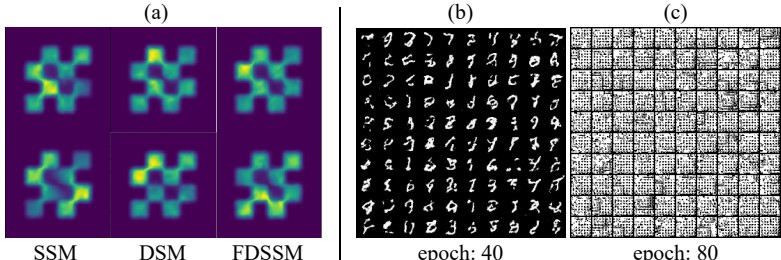

|  (a) | (b) | (c) |

SSM    DSM    FDSSM         epoch: 40        epoch: 80

Figure A5: (a) Visualized examples of EBFlow trained with SSM, DSM, and FDSSM on the Checkerboard dataset. (b) The samples generated by the Glow model at the 40-th training epoch. (c) The samples generated by the Glow model at the 80-th training epoch.

In addition, we observed that the sampling quality of EBFlow occasionally experiences a significant reduction during the training iterations. This phenomenon is illustrated in Fig. A5 (b) and (c), where the Glow model trained using our approach demonstrates a decline in performance with extended training periods. The underlying cause of this phenomenon remains unclear, and we consider it a potential avenue for future investigation.

