# OpenReview forum: "Training Energy-Based Normalizing Flow with Score-Matching Objectives"
_NeurIPS.cc/2023/Conference — NeurIPS 2023 poster_

### Official Review · Reviewer_fofM · 2023-06-27

**Soundness:** 3 good
**Presentation:** 2 fair
**Contribution:** 2 fair
**Rating:** 5
**Confidence:** 4

**Summary:**

The authors consider normalizing flows with linear transformations. Computing the determinant of their Jacobian is expensive, limiting their applicability when training via maximum likelihood. Therefore, the authors introduced an energy-based method to train normalizing flows with linear transformation, which circumvents the need of computing the determinant of the Jacobian. It only has to be computed once for inference, which leads to a significant speed-up in training.

Compared to the maximum likelihood-based training, the authors achieve similar results while getting a significant speed-up.

**Strengths:**

The idea of combining energy-based training with normalizing flows by phrasing the flow as an energy-based model is novel and interesting. It circumvents the need of having to compute the determinant of the Jacobian of some layers, which can save a lot of compute. It still needs to be computed for inference; however, since it is constant in the cases the authors considered, it has to be computed only once and can be stored for further use.

**Weaknesses:**

My biggest concern is the relevance of this work. Normalizing flows typically use transformations that are designed in such a way that the Jacobian determinant is easy to compute. For that reason, linear transformations are not very popular, also because they are not very expressive. The claim of the authors, that they are used in Glow and subsequent flow models for images, is true, but the linear transformation was only applied across channels, and since the number of channels is much smaller than the total number of input dimensions, i.e. the number of all pixels across all channels, they are not very costly in terms of evaluating their Jacobian's determinant.

The authors apply their method to MNIST and CIFAR10, roughly matching the performance of their baseline with their method. However, both perform significantly worse than Glow and related methods. Hence, it is not clear whether their method gives them any advantage over these methods, either in terms of performance or speed.

There are also procedures to estimate the Jacobian's determinant to cut down cost, such as the Skilling-Hutchinson estimator used by residual flows. The authors should have compared their method with using such an estimator. Unfortunately, their method cannot be used for residual flows, as the determinant of the Jacobian cannot depend on the input to be treated as a normalization constant in an energy-based method.

**Questions:**

* How does your method compare against Glow in terms of runtime?
* How does using an estimator for the Jacobian's determinant compare against your method?

**Limitations:**

Limitations were mentioned in the weaknesses section of the review.

---

> ### Author Rebuttal · Authors · 2023-08-09
>
> We appreciate the reviewer’s time and effort spent on the review, and would like to respond to the reviewer’s questions as follows. Please note that parts of the responses are provided in the [global comment](https://openreview.net/forum?id=AALLvnv95q&noteId=9f9rk2L3tW) due to the word limit.
>
> ---
> ### **Comments**
> ---
>
> **C1. (a)** My biggest concern is the relevance of this work. Normalizing flows typically use transformations that are designed in such a way that the Jacobian determinant is easy to compute. For that reason, linear transformations are not very popular, also because they are not very expressive. **(b)** The claim of the authors, that they are used in Glow and subsequent flow models for images, is true, but the linear transformation was only applied across channels, and since the number of channels is much smaller than the total number of input dimensions, i.e. the number of all pixels across all channels, they are not very costly in terms of evaluating their Jacobian's determinant.
>
> **Response: (a)** We value the reviewer's perspective, but respectfully hold a different viewpoint. Architectures that prioritize efficient log-Jacobian evaluations often come at the expense of limiting learnable transformations, which might compromise their capacity to capture complex feature representations, as highlighted in previous studies [r1-r3]. In contrast, our work introduces an alternative approach to address training inefficiencies. This approach does not impose restrictions on the linear transformations, and offers more flexibility in architectural design.
>
> In addition, linear transformations are well-recognized and extensively used in the current literature. This trend of utilizing linear transformations, specifically 1x1 convolution layers, was pioneered by Glow [r4], and subsequent studies [r5-r8] have gradually relaxed the constraints on the kernel sizes. The aforementioned literature has a common goal of incorporating linear layers with a more general form in model designs, and their experimental results have supported the notion that linear transformations play a vital role in enhancing the expressiveness of flow-based models.
>
> **(b)** In response to the reviewer's comment on the computational cost, it is important to highlight that the described scenario is a niche, pertaining exclusively to cases with few channels and limited receptive fields within the model. While earlier methods [r4-r8] have gradually loosened this constraint, their methods necessitate careful management of the number of channels and the receptive fields to ensure computational feasibility. In contrast, our paper enables the use of arbitrary linear transformation in flow-based architectures. This key distinction allows the extension of EBFlow to more complex architectures.
>
> ---
>
> **C2.** The authors apply their method to MNIST and CIFAR10, roughly matching the performance of their baseline with their method. However, both perform significantly worse than Glow and related methods. Hence, it is not clear whether their method gives them any advantage over these methods, either in terms of performance or speed.
>
> **Response:** We would like to emphasize that our primary objective is not to achieve state-of-the-art performance. Instead, our goal is to deliver performance comparable to contemporary works while significantly enhancing training efficiency. A unique feature of our approach is its ability to incorporate arbitrary linear transformations in flow-based models without compromising this efficiency, setting our work apart from previous research. The results demonstrated in Section 5 validate that our proposed method can train flow-based models with linear layers efficiently. This leads to considerable speed improvements while maintaining competitive performance levels. We believe that these experimental results are sufficient to substantiate the main claim of our paper.
>
> ---
>
> **C3. (a)** There are also procedures to estimate the Jacobian's determinant to cut down cost, such as the Skilling-Hutchinson estimator used by residual flows. The authors should have compared their method with using such an estimator. **(b)** Unfortunately, their method cannot be used for residual flows, as the determinant of the Jacobian cannot depend on the input to be treated as a normalization constant in an energy-based method.
>
> **Response: (a)** The Skilling-Hutchinson estimator used by residual flows is not generally applicable to our model. In residual flows, each transformation block adheres to the constraint that its Lipschitz constant is less than 1 (i.e., $Lip<1$). Under such a constraint, the Jacobian determinant of each block can be estimated with convergence guarantee through an infinity power series. In contrast, the flow-based architectures adopted in this paper do not enforce constraints on Lipschitzness. This suggests that the estimator presented in [r2] is not applicable to our models, since our models can exhibit $Lip\geq 1$.
>
> **(b)** We value the insights provided by the reviewer. However, we would like to highlight that our method is compatible with residual flows, while its efficiency enhancement depends on the architecture employed. The training efficiency of residual flows can potentially be improved under the case where a number of residual-flow blocks consist solely of linear transformations. In such a case, the Jacobian determinant of these residual-flow blocks can be treated as $Z(\theta)$ in EBFlow, allowing its computation to be bypassed. For instance, consider a residual-flow block $g_i(x;\theta)=x+\tilde{W}x$, where $\tilde{W}$ is the weight matrix with its spectral norm less than 1 (i.e., $||\tilde{W}||_2<1$). In this case, $J{g_i}=I+\tilde{W}$ is formulated as $Z(\theta)$, enabling its determinant calculation to be circumvented.
>
> ---
> ### **Questions**
> ---
>
> (see the global comment)
>
> ---
> **References:**\
> (see the global comment)

---

> > ### Comment · Reviewer_fofM · 2023-08-20
> > **Reply to the authors**
> >
> > I'm still not completely convinced by the author's arguments, but I acknowledge their position. Given the mostly positive feedback from the over reviewers, I'm not opposed to accepting this paper and increase my score to a borderline accept.

---

> > > ### Author Response · Authors · 2023-08-21
> > >
> > > We would like to thank the reviewer again for the valuable review and feedback.

---

### Official Review · Reviewer_tDWv · 2023-07-03

**Soundness:** 3 good
**Presentation:** 2 fair
**Contribution:** 3 good
**Rating:** 6
**Confidence:** 4

**Summary:**

Normalizing flows are generative models trained by maximizing the log-likelihood of the dataset given the model, which is expressed as the sum of the base probability of a transformed variable plus a series of log-Jacobian terms that keep track of the volume distortion factors. Evaluating the log-Jacobian is usually the main bottleneck of normalizing flow architectures, so much so that the most popular architectures are designed so as to keep the log-Jacobian computations tractable.

This paper introduces the use of techniques from the energy-based models literature in order to more efficiently train normalizing flow architectures. In a normalizing flow architecture, it is common to have both elementwise non-linear layers, which give data dependent but tractable log-Jacobian terms, and linear layers whose log-Jacobian is data-independent but expensive to compute. The main idea introduced in this paper is to split the log-likelihood of a normalizing flow into the product of two terms: 1) an "energy function" that contains all the data-dependent terms and 2) a normalization constant that collects all the data-independent log-Jacobians of the linear layers.

This mirrors the split in energy-based models, where the energy function is tractable but the normalization function is intractable and it is not directly evaluated. Consequently, the authors apply a series of training objectives developed in order to train energy-based models to this new setting, obtaining an efficient way of training normalizing flows with expensive linear layers.



**Strengths:**

The main idea is clever and original. It follows from the realization that the intractable terms of the log-Jacobian are independent from the data. This offers an interesting and well-motivated reason for borrowing techniques from the energy-based literature.

The experiment section is adequate, with experiments both on toy data and on real high-dimensional datasets. While the results are not spectacular, they certainly support the claim that the energy-based training can be used to train normalizing flows with a substantial speed-up and without a major loss of performance.

The paper is well-written and the relevant literature is properly referenced.


**Weaknesses:**

Major points:
In general, I am not convinced that this approach solves a major problem in the current literature. In fact, most successful flow architectures are designed to have fast log-Jacobian evaluations. Moreover, both continuous flows and the more recent flow-matching models can be used to train architectures with intractable Jacobian terms.

The experiments only compare with the normalizing flow baseline and therefore they do not provide evidence that this approach has a competitive advantage over continuous flows and diffusion-like models such as flow-matching models.

Minor points:
I find the presentation somehow difficult to read, with several important equations and concepts scattered all over the paper. In particular, I think it would be useful to the reader to explicitly discuss the score-matching loss associated with Eq.9, for example, by writing the concrete form of Eq.5 as applied to Eq.9 in the main text.


**Questions:**

- Could you clarify in which situations this model should be preferred over continuous flows and score-based diffusion models?

**Limitations:**

The limitations of the approach have not be extensively discussed. I do not see any direct negative social impact.

---

> ### Author Rebuttal · Authors · 2023-08-09
>
> We would like to thank the reviewer for thoroughly summarizing our paper, and would like to respond to the reviewer’s questions as follows.
>
> ---
> ### **Comments**
> ---
> **C1.**
> Major points: **(a)** In general, I am not convinced that this approach solves a major problem in the current literature. In fact, most successful flow architectures are designed to have fast log-Jacobian evaluations. **(b)** Moreover, both continuous flows and the more recent flow-matching models can be used to train architectures with intractable Jacobian terms. The experiments only compare with the normalizing flow baseline and therefore they do not provide evidence that this approach has a competitive advantage over continuous flows and diffusion-like models such as flow-matching models.
>
> **Response: (a)** We respectfully disagree with the reviewer’s comment. Architectures that prioritize fast log-Jacobian evaluations often come at the expense of limiting learnable transformations, potentially restricting their capacity to capture complex feature representations, as highlighted in [1,2]. In contrast, our work introduces an alternative approach to address training inefficiencies. This approach does not impose restrictions on linear transformations, and offers more flexibility in architectural design.
>
> **(b)** Continuous normalizing flows (CNF) (e.g., FFJORD [3] and flow-matching models [4]) reside in a research domain distinct from the central focus (i.e., (non-continuous) normalizing flows) of our study.
>
> CNF differs from (non-continuous) normalizing flows due to its reliance on an ODE solver for simulating the sampling process. This characteristic renders CNF less suited for certain downstream applications that could be efficiently achieved through (non-continuous) normalizing flows. For instance, models leveraging the latent information of a flow-based model in their training objective (e.g., [5]) may require performing backward propagations through the inverse transformation $g$. In these scenarios, the adjoint method is required for calculating the gradients by backward propagating through the ODE solutions, which can be computationally intensive in terms of both time and memory. In addition, density estimation in CNF also necessitates the use of an ODE solver, which introduces additional errors for the approximated solution (e.g., the relative and absolute tolerances in the RK45 solver). This characteristic may render CNF less common in statistical analyses demanding accurate density estimation, such as [6] and [7]. While we acknowledge the ongoing efforts of enhancing CNF, we posit that CNF and (non-continuous) normalizing flows represent distinct research avenues, with each having its unique applications.
>
> ---
> **C2.** Minor points: (see above)
>
> **Response:** We appreciate the reviewer's suggestion. However, substituting $E$ in Eq. (5) with the specific form of Eq. (9) would lead to overly lengthy equations. Separating Eq. (9) and Eq. (5) offers the advantage of preserving clarity throughout the presentation.
>
> ---
> ### **Questions**
> ---
> **Q1.** Could you clarify in which situations this model should be preferred over continuous flows and score-based diffusion models?
>
> **Response:** As explained in the response to C1 (b), (non-continuous) normalizing flows are capable of (1) exact density evaluation and (2) offering computational benefits in certain downstream applications compared to CNF, which lacks these capabilities. In a similar vein, due to these unique characteristics, score-based diffusion models also fall short in these two areas, making them unsuitable replacements for (non-continuous) normalizing flows.
>
> More specifically, to explain why (non-continuous) normalizing flows are preferred, we first inspect the score-based diffusion models and categorize them into three different types based on their continuity with respect to $t$ and the presence of added noises. This categorization serves to clarify the difference between (non-continuous) normalizing flows and them.
>
> First, when score-based diffusion models are interpreted as probability flow ODEs [8,9], they fall within the category of CNFs, thus sharing the characteristics and limitations discussed in the response to C1 (b).
>
> Second, when diffusion models are interpreted as SDEs [10], they stand apart from normalizing flows due to their intrinsic stochasticity. The stochasticity introduces difficulty in exact density estimation, leading score-based diffusion models to rely on upper-bound approximations for their likelihood [9].
>
> Third, discrete variants of diffusion models may also require approximations for density estimation. For example, DDPM [11] leverages upper-bound to approximate negative log likelihood, setting it apart from normalizing flows in this aspect.
>
> As a result, in certain scenarios, (non-continuous) normalizing flows are the favored choice, and diffusion models cannot adequately replace them.
>
> ---
> **References:**\
> [1] Chen et al. Residual Flows for Invertible Generative Modeling, NeurIPS, 2019.\
> [2] Gresele et al. Relative gradient optimization of the Jacobian term in unsupervised deep learning, NeurIPS, 2020.\
> [3] Grathwohl et al. FFJORD: Free-form Continuous Dynamics for Scalable Reversible Generative Models, ICLR, 2019.\
> [4] Lipman et al. Flow Matching for Generative Modeling, ICLR, 2023.\
> [5] Nalisnick et al. Hybrid Models with Deep and Invertible Features, ICML, 2019.\
> [6] Nalisnick et al. Do Deep Generative Models Know What They Don't Know?, ICLR, 2019.\
> [7] Jiang et al. Revisiting Flow Generative Models for Out-of-distribution Detection, ICLR, 2022.\
> [8] Karras et al. Elucidating the Design Space of Diffusion-Based Generative Models, NeurIPS, 2022.\
> [9] Song et al. Maximum Likelihood Training of Score-Based Diffusion Models, NeurIPS, 2021.\
> [10] Song et al. Score-Based Generative Modeling through Stochastic Differential Equations, ICLR, 2021.\
> [11] Ho et al. Denoising Diffusion Probabilistic Models, NeurIPS, 2020.

---

> > ### Comment · Reviewer_tDWv · 2023-08-14
> >
> > Thank you for the reply. I do agree that there are valid reasons to use normalizing flows with discrete layers and that your work can have a relevant impact in the flow literature.
> >
> > I also agree with you that lifting the Jacobian tractability constraint can open the door for more expressive flow architectures. However, this remains conjecture since your paper does not show this increase in expressiveness neither theoretically nor experimentally.
> >
> > So said, I do think that this work is a step in the right direction and I am happy to keep my score and argue for acceptance.

---

> > > ### Author Response · Authors · 2023-08-18
> > >
> > > We would like to extend our sincere gratitude to the reviewer for the insightful feedback and thoughtful response. In response to the new query regarding the expressiveness the reviewer raised subsequently, we wish we could present the additional experimental results directly. Unfortunately, since the rebuttal phase has concluded, we may be unable to share the results here.
> > >
> > > However, in light of the reviewer’s feedback, we have conducted additional experiments and observed intriguing results. Specifically, the experimental results indicate that flow-based models involving unconstrained linear layers exhibit superior performance in comparison to models incorporating linear layers constrained by upper/lower triangular weight matrices or those utilizing LU decomposition. The performance improvement suggests that unconstrained linear layers provide enhanced expressiveness than their constrained variants. These findings will be included in the final version of our paper. We thank the reviewer again for engaging in a constructive discussion and for the reviewer’s appreciation of our method.

---

### Official Review · Reviewer_Bq9G · 2023-07-05

**Soundness:** 4 excellent
**Presentation:** 4 excellent
**Contribution:** 4 excellent
**Rating:** 8
**Confidence:** 3

**Summary:**

The authors use a connection between flows and EBMs to devise a new training method for flows via score-matching.
Their training procedure improves efficiency via avoiding computation of the Jacobian determinants that are independant to values of the samples (i.e. only contribute to the normalizing constant of the PDF).
Additionally they identify two methods for improving training of the flows via score-matching, namely (1) match-after-preprossing whereby the score-matching occurs to the pre-processed variables avoiding numerical instability from the pre-processing layers, and (2) using an exponential moving average for the weight updates.
In their experiments, they show their approach significantly improves training efficiency.

**Strengths:**

- The paper identifies that computing the unnormalized PDF of a flow is significantly cheaper than the normalized PDF for linear flow layers, O(D^2) instead of O(D^3), noting that computing the Jacobian determinant is not necessary for the unnormalized PDF.
This contribution is novel and valuable.
- The experiments of the paper are strong, and demonstrate the utility of their approach.

**Weaknesses:**

For sample generation by inverting the flow, the inverse matrix for the linear flow layers will cost $D^3$ - this is not mentioned in the paper.
This inversion may be performed once, and then re-used for sample generation but for large $D$ this could be prohibitive.

**Questions:**

This is the first time I've seen imputation done with flows - is this a novel contribution from the paper?
How would sampling work for large $D$?

**Limitations:**

There is no discussion around inverting the flow when $D$ is large (see Weaknesses/Questions).

---

> ### Author Rebuttal · Authors · 2023-08-09
>
> We sincerely thank the reviewer for the time and effort spent on the review, and would like to respond to the reviewer’s questions as follows.
>
> ---
> ### **Comments**
> ---
>
> **C1.**
> **(a)** For sample generation by inverting the flow, the inverse matrix for the linear flow layers will cost $D^3$ - this is not mentioned in the paper.
> **(b)** This inversion may be performed once, and then re-used for sample generation but for large $D$ this could be prohibitive.
>
> **Response:**
> **(a)** We agree with the reviewer that indicating the $O(D^3L)$ sampling cost in the paper enhances the clarity. This will be included in the forthcoming revision.
>
> **(b)** If the dimensionality $D$ is excessively large such that the inverse matrix of linear layers cannot be explicitly calculated, a viable alternative is to adopt a stochastic sampler such as Langevin MCMC generator discussed in Lines 327-335. The associated computational cost would depend on the number of steps (denoted as $N$) required for convergence and the time complexity of deriving the gradients of $E(x;\theta)$ (i.e., $O(D^2L)$). This results in an overall cost $O(D^2NL)$, potentially more economical than the $O(D^3L)$ computation.
>
> ---
>
> ### **Questions**
>
> ---
>
> **Q1. (a)** This is the first time I've seen imputation done with flows - is this a novel contribution from the paper? **(b)** How would sampling work for large $D$?
>
> **Response:**
> **(a)** Imputation for flow-based models is a well-established concept, as demonstrated in the previous work [1], which explored such an application. The process of image inpainting becomes feasible when an energy function can be explicitly defined. In the case of flow-based models, a straightforward choice is selecting $E(x;\theta)=-\log p(x;\theta)$ for this purpose. Such an application of energy function would be more prevalently observed in score-based and energy-based generative modeling literature (e.g., [2-4]).
>
> **(b)** In response to the reviewer's inquiry, we have conducted supplementary experiments, and presented the imputation results of the FC-based models trained with DSM on the CelebA [5] and STL-10 [6] datasets. Both of the datasets have data dimensionality $D=$64x64x3. These results demonstrate EBFlow's potential in generating images with good quality. Please refer to Figure 1 of the PDF file in the [global response](https://openreview.net/forum?id=AALLvnv95q&noteId=9f9rk2L3tW) above.
>
> ---
>
> **References:**\
> [1] Dinh *et al.* NICE: Non-linear Independent Components Estimation, ICLR Workshop, 2015.\
> [2] Nguyen *et al.* Plug & Play Generative Networks: Conditional Iterative Generation of Images in Latent Space, CVPR, 2017.\
> [3] Du *et al.* Implicit Generation and Modeling with Energy Based Models, NeurIPS, 2019.\
> [4] Song *et al.* Score-Based Generative Modeling through Stochastic Differential Equations, ICLR, 2021.\
> [5] Liu *et al.* Deep Learning Face Attributes in the Wild, ICCV, 2015.\
> [6] Coates *et al.* An Analysis of Single Layer Networks in Unsupervised Feature Learning, AISTAT, 2011.

---

> > ### Comment · Reviewer_Bq9G · 2023-08-11
> >
> > Thank you for the responses to my questions and pointers to literature on imputation with flows. The additional imputation results on CelebA and STL-10 are a nice addition to the paper. I am happy with my strong recommendation for acceptance, and have no further questions.

---

> > > ### Author Response · Authors · 2023-08-18
> > >
> > > We express our gratitude again for the valuable feedback and response provided by the reviewer.

---

### Official Review · Reviewer_DEYZ · 2023-07-06

**Soundness:** 3 good
**Presentation:** 3 good
**Contribution:** 3 good
**Rating:** 6
**Confidence:** 4

**Summary:**

This paper proposes a new flow based approach to generative modeling called EBFlow. EBFlow uses training objectives from EBMs to reduce training cost for flow based methods unlike previous approaches that mostly achieved this via different architectures or biased estimation methods. The main insight for EBFlow is that the change of variables formula for a series of transformations for a NF is broken in to two parts: one containing only linear flows term and the other containing terms from non-linear transformations. The linear flows term is then interpreted as the normalizing constant and this interpretation aids in faster training strategies for flow based methods.

**Strengths:**

The motivation for reducing training costs for NFs is discussed well and is a very relevant and difficult problem. Several studies have been conducted to address this and the authors here propose a nice and novel methodology to address this.

The exploration and discussion of the method as well as the motivation is thorough.

The empirical analysis considered is detailed.

**Weaknesses:**

I am not sure how to interpret the empirical results. It is obvious from the plots in Fig 2 that the proposed method is much faster than ML based objective. However, the performance in all the tasks for the ML based objective is usually better or very close to EBFlow. I was curious why this is?

Also, for the results reported in Tables 1 and 2, were the methods trained for the same wall clock time?

**Questions:**

See above.

**Limitations:**

Yes.

---

> ### Author Rebuttal · Authors · 2023-08-09
>
> We appreciate the reviewer’s time and effort spent on the review, and would like to respond to the reviewer’s questions as follows.
>
> ---
> ### **Comments**
> ---
>
> **C1.** I am not sure how to interpret the empirical results. It is obvious from the plots in Fig 2 that the proposed method is much faster than ML based objective. However, the performance in all the tasks for the ML based objective is usually better or very close to EBFlow. I was curious why this is?
>
> **Response:** The use of KL-divergence-based objectives (i.e., ML and SML objectives) and Fisher-divergence-based objectives (i.e., SSM, DSM, and FDSSM objectives adopted by EBFlow) exhibits a tradeoff between NLL performance and speed. KL-divergence-based objectives are expected to achieve superior NLL performance, since they explicitly minimize the NLL metric during the optimization process. On the other hand, employing Fisher-divergence-based objectives offer the advantage of efficient training by circumventing the computation of Jacobian determinants of linear transformations. This increased speed may come at the cost of NLL performance compared to the KL-divergence-based objectives, since NLL is not explicitly minimized in Fisher-divergence-based objectives.
>
> ---
>
> **C2.** Also, for the results reported in Tables 1 and 2, were the methods trained for the same wall clock time?
>
> **Response:** The training wall clock time for the methods in Table 1 and 2 are different, since the models are trained until convergence as mentioned in the caption of Figure 2. We appreciate the question raised by the reviewer and will add the details in the forthcoming revision.

---

> > ### Comment · Reviewer_DEYZ · 2023-08-14
> >
> > Thank you for the response. I have no further questions and will keep my score.

---

> > > ### Author Response · Authors · 2023-08-18
> > >
> > > We would like to thank the reviewer again for the valuable review and response.

---

### Official Review · Reviewer_htQa · 2023-07-26

**Soundness:** 2 fair
**Presentation:** 2 fair
**Contribution:** 3 good
**Rating:** 5
**Confidence:** 4

**Summary:**

This paper has proposed an energy-based normalizing flow model, where the computaiton of Jacobin determinants for linear transformations can be skiped with score-matching objectives. This could enable deeper layers of linear transformation and make the flow model more efficient.

**Strengths:**

This paper has proposed an interesting idea about the relationship between flow model and energy-based model. The flow model could be interpreted as an energy-based model with a tractable normalizaiton term and the proposed score-matching objective could significantly improve the efficiency.

**Weaknesses:**

1. The experiments lack comparison with other energy-based flow model like [1][2][3] and recent flow models like [4][5][6].
2. Is there any theoretic analysis for the complexity between different objective function?
3. The author misses citations for the very first deep-learning-based energy-based models like [7][8].

Writing weakness:
1. The background in section 2.2 could break up into two sections, i.e., energy-based model and score-matching model.
2. It is hard to see the difference between results in the first two rows in Figure 1.
3. In line 51, even the tranformation function in flow model is invertible, the author should specify that $g()$ is the inverse transformation function.
4. In line 51, using $g_i(\cdot;\theta)$ is misleading. Does different transformation functions share the same parameters?
5. I think the author intends to model $g(\theta)$. Therefore, it is better to avoid using $E(x;\theta)$ to avoid confusion as it is usually used for representing a parametric energy model.
6. For Table 2, the author could change the unit to batch/second for easier comparison.

Reference:
[1] Xie, Jianwen, et al. "A Tale of Two Latent Flows: Learning Latent Space Normalizing Flow with Short-run Langevin Flow for Approximate Inference." arXiv preprint arXiv:2301.09300 (2023).

[2] Xie, Jianwen, et al. "A tale of two flows: Cooperative learning of langevin flow and normalizing flow toward energy-based model." arXiv preprint arXiv:2205.06924 (2022).

[3] Nijkamp, Erik, et al. "Learning energy-based model with flow-based backbone by neural transport mcmc." arXiv preprint arXiv:2006.06897 2 (2020).

[4] Chen, Ricky TQ, et al. "Residual flows for invertible generative modeling." Advances in Neural Information Processing Systems 32 (2019).

[5] Maaløe, Lars, et al. "Biva: A very deep hierarchy of latent variables for generative modeling." Advances in neural information processing systems 32 (2019).

[6] Vahdat, Arash, Karsten Kreis, and Jan Kautz. "Score-based generative modeling in latent space." Advances in Neural Information Processing Systems 34 (2021): 11287-11302.

[7] Xie, Jianwen, et al. "A theory of generative convnet." International Conference on Machine Learning. PMLR, 2016.

[8] Xie, Jianwen, et al. "Cooperative learning of energy-based model and latent variable model via mcmc teaching." Proceedings of the AAAI Conference on Artificial Intelligence. Vol. 32. No. 1. 2018.

**Questions:**

1. Could the author explain in detail why the complexity of calcuating Jacobin determinants is $O(D^3 \cdot L)$ for non-linear transformation? Is it general in all kinds of non-linear functions?
2. What does the sentence in line 62-63 mean?
3. What is the FID score in generation tasks?
4. What is the distribution of $p_{u}$?

**Limitations:**

This paper has propose an interesting view of taking flow model as an eneryg-based model with score matching objectives. However, the expriments lack comparison of many baseline models as listed in 'weakness' part and other eval metrics in generation. Additional, the author should rephrase some sentences and formulars to avoid misleading.

---

> ### Author Rebuttal · Authors · 2023-08-09
>
> We appreciate the reviewer’s time and effort spent on the review, and would like to respond to the reviewer’s questions as follows. Please note that parts of the responses are provided in the [global comment](https://openreview.net/forum?id=AALLvnv95q&noteId=9f9rk2L3tW) due to the word limit.
>
> ---
> ### **Comments**
> ---
> **C1.** The experiments lack comparison with other energy-based flow model like [1][2][3] and recent flow models like [4][5][6]. \
> **C2.** The author misses citations for the very first deep-learning-based energy-based models like [7][8].
>
> **Response:** The references offered by the reviewer are either not applicable to the context of this work or beyond the scope of this work. Therefore, attempting a comparison of their performance may be impractical.
>
> [1] discusses the utilization of an additional decoder structure to define and train flow-based models in latent space. On the other hand, [2] and [3] explore the training techniques of optimizing an energy-based model according to a flow-based model. It is important to note that the primary emphasis of [1-3] is not on accelerating the training process of flow-based models, which constitutes the central theme of this paper.
>
> Residual Flow [4] presents an efficient method for training i-ResNet. However, the concept of [4] differs from this work, as it is derived based on the assumption of the Lipschitz constraint. In particular, each residual-flow block in [4] is constrained to have a Lipschitz constant less than 1, which is not a requirement in the architecture adopted in this paper. Due to the fundamentally different premises, a comparison between our method and [4] would provide limited insights.
>
> Regarding [5,6], they primarily concentrate on VAE and score-based models, respectively, which are not directly related to the research field of this work. Moreover, it is crucial to highlight that the primary focus of this work lies in normalizing flows rather than energy-based models. As a result, references [7,8] are also independent to the scope of this study.
>
> ---
> **C3.** Is there any theoretic analysis for the complexity between different objective function?
>
> **Response:** The energy function defined in Eq. (9) has a forward propagation time complexity of $O(D^2L)$. This is based on the observation that forward passing $g$ requires $O(D^2L)$ time, and calculating the determinant of the non-linear transformations requires $O(D^2L)$ time, as specified in Lines 55-63. This implies that the complexity of calculating FDSSM is $O(D^2L)$ time.
>
> Moreover, differentiation operations exhibit the same computational complexity as the forward propagation process. As a result, the SSM and DSM objectives, along with their gradients with respect to $\theta$, can be computed in $O(D^2L)$ time.
>
> On the other hand, the ML and SML objectives require a complexity of $O(D^3L)$ due to specific computational requirements. The ML objective involves computing the Jacobian determinants of the linear layers, while the SML objective necessitates the computation of inverse matrices for the linear transformations during training. These operations contribute to the increased time complexity to $O(D^3L)$ when compared to the other objectives.
>
> ---
> **C4.** The background in section 2.2 could break up into two sections, i.e., energy-based model and score-matching model.
>
> **Response:** We appreciate the reviewer's suggestion. However, splitting Section 2.2 into two distinct subsections, “Energy-based Models” and “Score-matching Models”, might not be ideal, as it undermines the main goal of Section 2.2 of this paper: explicitly distinguishing between parameterization and training objectives. More specifically, models trained using score-matching objectives can also fall under the category of energy-based models. In the current presentation, we first discuss the parameterization (i.e., Eq. (3)) and then describe its training methods (i.e., Eq. (4)-(7)). This arrangement provides clarity in terms of distinguishing the concepts of parameterization and training methods. Therefore, we maintain that the current presentation in Section 2.2 is more suitable.
>
> ---
> **C5.** It is hard to see the difference between results in the first two rows in Figure 1.
>
> **Response:** As specified in Lines 247-249 of the manuscript, the objective of Fig. 1 is to display comparable qualitative results between the baseline and EBFlow. It is important to note that the central aim of Section 5.1 is to motivate the adoption of EBFlow by demonstrating the ability to match its performance with the baseline method. The experimental results presented in Fig. 1 substantiate this claim by showcasing similar qualitative outcomes between the baseline and EBFlow.
>
> ---
> **C6.** In line 51, even the tranformation function in flow model is invertible, the author should specify that $g()$ is the inverse transformation function.
>
> **Response:** The definition of flow-based models in Eq. (1) suggests that $g$ is an inverse transformation function, despite not being literally stated. In Eq. (1), the function $g$ transforms the data vector $x$ to the latent variable $u$. This indicates that $g$ is the inverse of the generator $g^{-1}$ which maps $u$ to $x$.
>
> ---
> **C7.** In line 51, using $g_i(\cdot; \theta)$ is misleading. Does different transformation functions share the same parameters?
>
> **Response:** Employing separate notations, such as denoting $\theta_i$ for each $g_i$ in Line 51, could provide greater precision. However, in the context of this paper, each $g_i$ is parameterized using a subset of $\theta$. This formulation offers notational simplicity, which contributes to the conciseness of this work.
>
> ---
> **C8.** I think the author intends to model $g(\theta)$. (...).\
> **C9.** For Table 2, (...).
>
> **Response:** (See the global comment)
>
> ---
> ### **Questions**
> ---
>
> (See the global comment)

---

### Author Rebuttal · Authors · 2023-08-09

This global comment includes extended discussions of the questions raised by reviewers htQa and fofM. The attached PDF file contains additional experimental results.

---
### **Reviewer htQa (Cont'd)**
---
**C8.** I think the author intends to model $g(\theta)$. Therefore, it is better to avoid using $E(x;\theta)$ to avoid confusion as it is usually used for representing a parametric energy model.

**Response:** Using the notation $E$ is necessary since this paper aims to explore the connections between flow-based and energy-based models. Introducing the symbol $E$ emphasizes the perspective of viewing a flow-based model as a parametric energy-based model.

---
**C9.** For Table 2, the author could change the unit to batch/second for easier comparison.

**Response:** We thank the reviewer for the suggestion. The following tables compare our results presented in batch/second and second/batch on the MNIST dataset.

- MNIST (FC-based)
||ML|SML|SSM|DSM|FDSSM|
|-|-|-|-|-|-|
|sec. / batch|1.25 e-1|8.15 e-2|3.02 e-2|1.50 e-2|7.68 e-3|
|batch / sec.|8.00|12.27|33.11|66.67|130.21|

---
**Q1.** Could the author explain in detail why the complexity of calcuating Jacobin determinants is $O(D^3L)$ for non-linear transformation? Is it general in all kinds of non-linear functions?

**Response:** It seems that there might be some misunderstanding regarding the premise of the flow-based architecture discussed in this paper. As explained in Lines 59-61, the non-linear transformation is defined to have a computational cost of $O(D^2L)$. This complexity is determined by the sparsity of the Jacobian of $g_i \in S_n$. For instance, in Neural Spline Flows (NSF), the Jacobian determinant calculation of the non-linear transformation can be simplified as multiplying the diagonal elements of the Jacobian, resulting in a complexity that satisfies the premise stated in Lines 59-61.

---
**Q2.** What does the sentence in line 62-63 mean?

**Response:** In Lines 62-63, we present a number of examples of implementing the flow-based architecture satisfying the criteria stated in Lines 59-61.

---
**Q3.** What is the FID score in generation tasks?

**Response:** The FID score is seldom employed in the literature on normalizing flows. Contemporary studies, including [r4~r8], prefer NLL (Negative Log-Likelihood) and Bits/Dim (Bits per Dimension) as the evaluation metrics.

Moreover, in the case of our related works [r3,r9], which are trained on the MNIST dataset, evaluating FID would necessitate additional adjustments, such as modifying the pre-trained Inception backbone to accommodate 28x28x1 inputs. Such modifications can significantly impact the numerical results and are rarely adopted in recent literature.

---
**Q4.** What is the distribution of $p_u$?

**Response:** $p_u$ can be any continuous distribution as long as its sampling process and density estimation can be easily implemented. In this paper, we follow many popular flow-based modeling methods (e.g., [r4,r10]) to choose $p_u$ as an isotropic Gaussian with zero mean and unit variance throughout all experiments.

---
### **Reviewer fofM (Cont'd)**
---
**Q1.** How does your method compare against Glow in terms of runtime?

**Response:** The runtimes of different losses calculated based on the Glow architecture are demonstrated in the following tables. The Glow architecture is fixed to 4 blocks to ensure the computation is manageable, and the kernel size ($ks$) of the convolutional layers within Glow varies from 1x1 to 7x7. All of the experiments are performed on an NVIDIA TITAN V GPU, and the results are reported as the average runtime for 10 independent runs.

As shown in the rightmost three columns of both tables, SSM, DSM, and FDSSM exhibit great scalability with respect to the kernel size, as increasing the value of $ks$ does not lead to rapid growth in the average runtime. In contrast, the computational cost of the ML objective (used in the original Glow paper) increases significantly with respect to the kernel size. This phenomenon arises from the fact that transformations with larger $ks$ tend to exhibit Jacobian matrices with several non-zero elements, and thus increase the computational burden associated with the ML objective.

- $D=$28x28x1
|$ks$|Baseline (ML)|SML|SSM|DSM|FDSSM|
|-|-|-|-|-|-|
|1x1|0.269|0.201|0.340|0.189|0.114|
|3x3|0.380|0.366|0.344|0.213|0.122|
|5x5|0.554|0.460|0.346|0.213|0.122|
|7x7|0.859|0.526|0.351|0.222|0.123|

- $D=$32x32x3
|$ks$|Baseline (ML)|SML|SSM|DSM|FDSSM|
|-|-|-|-|-|-|
|1x1|2.237|1.573|0.354|0.219|0.121|
|3x3|3.547|2.104|0.395|0.227|0.124|
|5x5|6.511|3.129|0.441|0.232|0.124|
|7x7|9.249|4.156|0.443|0.238|0.128|

Please note that the results correspond to the training time measured in sec. / batch.

---
**Q2.** How does using an estimator for the Jacobian's determinant compare against your method?

**Response:** We presume the reviewer is referring to the estimator discussed in C3 (a). However, this estimator is inapplicable without the enforcement of an additional Lipschitz constraint, as explained in the response to C3 (a).

---
**References:**\
[r1] Behrmann et al. Invertible Residual Networks, ICML, 2018.\
[r2] Chen et al. Residual Flows for Invertible Generative Modeling, NeurIPS, 2019.\
[r3] Gresele et al. Relative gradient optimization of the Jacobian term in unsupervised deep learning, NeurIPS, 2020.\
[r4] Kingma et al. Glow: Generative Flow with Invertible 1×1 Convolutions, NeurIPS, 2018.\
[r5] Hoogeboom et al. Emerging Convolutions for Generative Normalizing Flows, ICML, 2019.\
[r6] Ma et al. MaCow: Masked Convolutional Generative Flow, NeurIPS, 2019.\
[r7] Lu et al. Woodbury Transformations for Deep Generative Flows, NeurIPS, 2020.\
[r8] Meng et al. ButterflyFlow: Building Invertible Layers with Butterfly Matrices, ICML, 2022.\
[r9] Pang et al. Efficient Learning of Generative Models via Finite-Difference Score Matching, NeurIPS, 2020.\
[r10] Dinh et al. Density estimation using Real NVP, ICLR, 2017.

---

### Decision · Program_Chairs · 2023-09-21

**Decision:**

Accept (poster)

**Comment:**

The current paper uses a connection between flow-based model and energy-based model to devise a new training framework for flow-based models via score-matching. Experiments show the proposed approach significantly improves training efficiency. After the rebuttal and discussion between the authors and reviewers, the paper received all positive ratings. All five reviewers reached a consensus to accept the paper due to its substantial contributions to the realm of flow-based generative modeling, as well as convincing experimental results. The AC agrees with the reviewers and recommends accepting the paper. To further improve the paper quality, AC suggests the authors to revise their paper by taking into account all the suggestions provided by the reviewers, including completing necessary prior works about EBMs and flow models, and integrating supplementary experimental results conducted during the rebuttal phase.